# A CONSISTENT PATTERN FOR IDENTIFYING DECISIVE CODE SNIPPETS FOR LLM-BASED CODE INFERENCE

## ABSTRACT

Which parts of *pre-target input*[1] are most influential for next-token prediction in the context of programming languages? In this paper, we present evidence that code snippets at specific locations in pre-target inputs play a decisive role in large language model (LLM) inference, and these snippets exhibit a consistent pattern. Firstly, we introduce a novel *causal tracing method* to identify tokens, so-called *high-information tokens*, that significantly contribute to next-token prediction. Building on this, we propose a *multi-phase causal tracing process* to analyze the importance distribution of high-information tokens, revealing a consistent pattern, named the *Important Position Rule* (IPR). To further validate this **hypothesis**, we assess the role of IPR across various LLMs, languages, and tasks. Our extensive evaluations for code translation, code correction and code completion tasks (Java, Python, C++) on models CodeLlama-7b/13b/34b-Instruct (Roziere et al., 2023) and gpt-3.5/4-turbo (Ouyang et al., 2022), confirm this hypothesis. Furthermore, we observe that IPR exhibits structural and semantic properties similar to the ⟨subject, relation, object⟩ paradigm in natural language. Leveraging this insight, we successfully combine IPR with the knowledge editing method ROME (Meng et al., 2023) in order to repair translation errors, achieving a correction rate of 62.73% to 75.31%. To our knowledge, this is the first application of knowledge editing in the context of programming languages.

## 1 INTRODUCTION

Recently, researchers have delved into the internal mechanisms of large language models. The inherent syntactic structures in natural language, such as ⟨subject, relation, object⟩, provide solid support for interpretable inference (Kim et al., 2024; Stolfo et al., 2023; Katz et al., 2024; Haviv et al., 2023) and knowledge editing (Meng et al., 2022; Zhang et al., 2024; Meng et al., 2023; Gupta et al., 2024). However, the existence of analogous mechanisms in code-based LLMs remains uncertain, which poses challenges for the aforementioned research in programming language contexts. In this paper, we demonstrate that code snippets located at specific positions play a crucial role in guiding LLM-based code inference, and that these snippets exhibit a consistent pattern.

In the field of natural language processing (NLP), GPT models have exhibited a remarkable ability to learn and utilize syntactic structures, which are essential for establishing internal correlations between words (Petroni et al., 2019; Rai et al., 2024; Bajpai et al., 2024). For example, given the prompt "*The Eiffel Tower is located in*", GPT can accurately predict "*Paris*". It underscores the importance of syntactic structures in guiding inference and enhancing predictions (Mikolov et al., 2013; Touvron et al., 2023; Ouyang et al., 2022; Roziere et al., 2023). This leads us to the **Core Research Question**: Is there a consistent pattern that significantly contributes to LLM-based code inference? To address this question, we first quantify the information content of individual tokens to assess their contribution to the inference process. Subsequently, we discover the importance distribution pattern of high-information tokens and examine how this pattern influences the inference process, ultimately providing new insights into the inner workings of LLM-based code inference.

Existing tracing methods primarily focus on perturbing the training or test datasets to identify high-information tokens based on changes in generated tokens. However, relying solely on target token

---

[1]Pre-target input: It refers to the combination of an input sequence and a generated output prefix, which together provide the contextual basis for generating a target token.

changes to determine the impact of input tokens is a rather crude approach (Dong et al., 2019; Brown et al., 2020; Devlin et al., 2018). On the one hand, the output of the Transformer model is a probability distribution over multiple tokens in the vocabulary. Therefore, this approach overlooks other input tokens that have a relatively high influence (Adler et al., 2016; Hao et al., 2021; Geva et al., 2022). On the other hand, the generation of the target token is determined by both the source input sequence and the generated output prefix (i.e., pre-target input), yet existing research typically perturbs only the input sequence while neglecting the latter (Liu et al., 2023; Cho et al., 2014). In this paper, we propose introducing perturbations to the tokens in both the input sequence and the generated prefix. By analyzing the resulting fluctuations in the internal representations, we can effectively quantify the information content of individual tokens.

In recent years, knowledge editing has garnered widespread attention in the field of natural language processing (Mazzia et al., 2023; Wei et al., 2023). Meng et al. (2023) highlights that fine-tuning the middle layer weights of GPT models enables them to rapidly learn new knowledge, such as "*The Eiffel Tower is located in Berlin*". We argue that the success of knowledge editing stems from the properties of the latent space. After extensive training, the frequent ⟨subject, relation, object⟩ structures in the training samples establish a robust association between the tokens "*Eiffel Tower*" and "*Paris*". Therefore, when editing at the position of "*Eiffel Tower*", effective information substitution can be achieved. However, existing knowledge editing techniques primarily rely on the ⟨subject, relation, object⟩ structure, which is typically limited to the natural language. In this paper, we find that code snippets identified by IPR in the input sequence and the output prefix exhibit strong semantic and syntactic correlations, similar to the close relationship between phrases of "subject" and "object" in natural language. Building on this observation, we apply IPR in knowledge editing to rectify errors in Java to Python translation. The main contributions of this paper are the following:

- We propose a causal tracing method that interacts the low-dimensional text sequence with the high-dimensional internal representation. This approach quantifies the information content of individual tokens in both the input sequence and the generated output prefix, enabling an assessment of each token's contribution to the next-token prediction.

- We introduce a multi-phase causal tracing process, revealing a consistent pattern of high-information tokens, named the important position rule.

- We validate the role of IPR in code inference across diverse models, tasks, and programming languages, including code translation, code correction, and code completion, utilizing CodeLlama-7b/13b/34b-Instruct and gpt-3.5/4-turbo with Java, Python, and C++. Our evaluation shows that code snippets identified by IPR play a critical role in next-token prediction. We also confirmed their robust generalization capabilities, providing valuable interpretability for LLM-based code inference.

- We combine IPR with ROME in the context of programming languages, generalizing this knowledge editing beyond the NLP context. Our approach effectively corrected errors in Java to Python translation, with a correction rate of 62.73% to 75.31%.

## 2 RELATED WORK

Currently, mainstream methods for interpreting large language models include causal mediation analysis (Hicks & Tingley, 2011; Imai et al., 2010), influence function (Cook & Weisberg, 1980), knowledge attribution (Powell et al., 2015; Bricker, 2020), and counterfactual analysis (Keohane, 2009; Hernán & Robins, 2010), etc.

One straightforward yet effective method for interpreting LLMs involves locally perturbing inputs. This approach allows for a detailed analysis of which components of the input sequence are most influential in guiding the model's predictions (Lundberg & Lee, 2017; Wiegreffe & Pinter, 2019a; Ribeiro et al., 2016). Li et al. (2023) introduced a selective context method, which enhances the LLM inference efficiency by identifying and pruning redundancies in the input context, resulting in a more concise input. Parallel to these efforts, Jiang et al. (2023) proposed a coarse-to-fine prompt compression method, LLMLingua, which effectively captures the interdependencies among compressed content. From a causal perspective, Feder et al. (2021a) introduced CausaLM, a framework that utilizes counterfactual language to produce interpretations of causal models (Simon & Rescher,

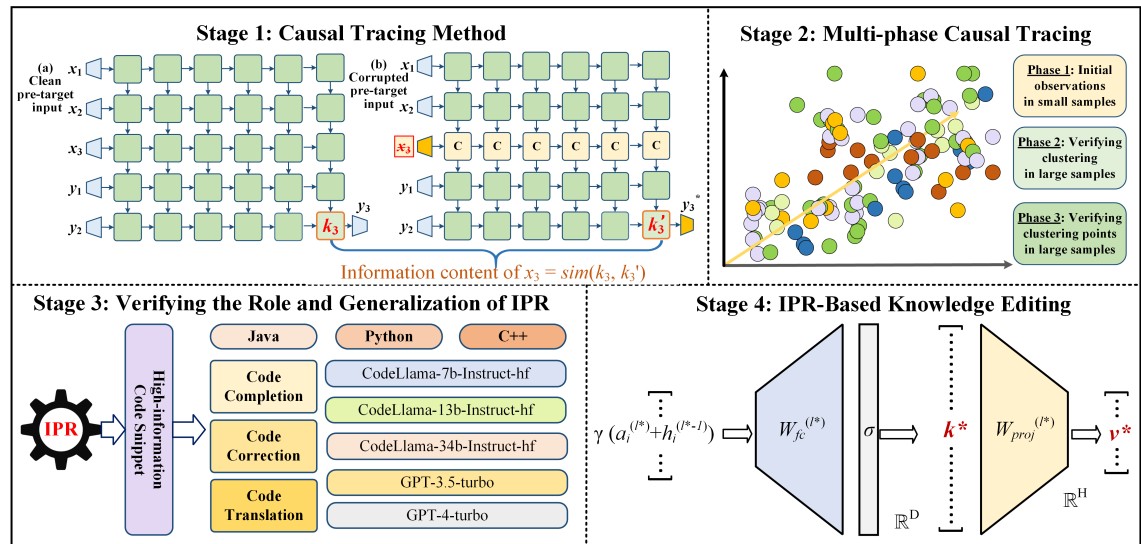

Figure 1: **Exploration and Application Workflow of the Important Position Rule.** First, we employed the causal tracing method and a multi-phase causal tracing process to investigate the importance distribution pattern of high-information tokens, leading to the IPR hypothesis. Next, we validate the role and generalization of IPR across various programming languages, LLMs, and code tasks. Finally, we integrate IPR with ROME, successfully performing knowledge editing in the code-based LLM. After thorough evaluation, IPR demonstrates significant potential in enhancing interpretable inference and facilitating knowledge editing in the context of programming languages.

1966; Hernán & Robins, 2010; Feder et al., 2021b). Then, Kramár et al. (2024) proposed the attribution patching method, which employs linear approximation at the corrupted prompt to evaluate the impact of local changes in the model. To identify which training points contribute to the specific prediction, Koh & Liang (2017) leveraged the influence function (Hampel, 1974) to trace model predictions from a robustness perspective, revealing insights about how models rely on and infer from training data. Building on this, Geva et al. (2022) presented LM-Debugger, providing a granular explanation of the model's internal prediction processes (Wallace et al., 2019). By observing the effects of erasing components of the representation (e.g. input word vector dimensions, intermediate hidden units, or input words), Li et al. (2017) analysed and explained the decisions of the neural model. Furthermore, some researchers have investigated the role of intermediate representations from attention modules in explaining model predictions. Wiegreffe & Pinter (2019b) introduced four tests to assess when/whether attention can serve as an explanation, providing insights into model reasoning. Following this, Wu et al. (2021) introduced a parameter-free probing technique for analyzing pre-trained language models that eliminates the need for direct supervision and avoids incorporating extra parameters during the probing process.

The central idea in existing research is to perturb input sequences and observe changes in the generated tokens, thereby identifying correlated internal components, input tokens, or training samples relevant to the model predictions (Lundberg & Lee, 2017; Wiegreffe & Pinter, 2019a; Dai et al., 2022; Ribeiro et al., 2016). However, since the output of Transformer models is a probability distribution over multiple tokens, relying solely on changes in the target token is a crude approach that risks overlooking other input tokens with relatively higher influence. Moreover, the output of target tokens is dominated by both the source input sequence and the generated output prefix, yet existing studies have neglected the influence of the output prefix.

## 3 TRACING IMPACTFUL INFORMATION SOURCE FOR CODE INFERENCE

In this section, we combine the causal tracing method with the multi-phase causal tracing process, revealing a consistent pattern among high-information tokens.

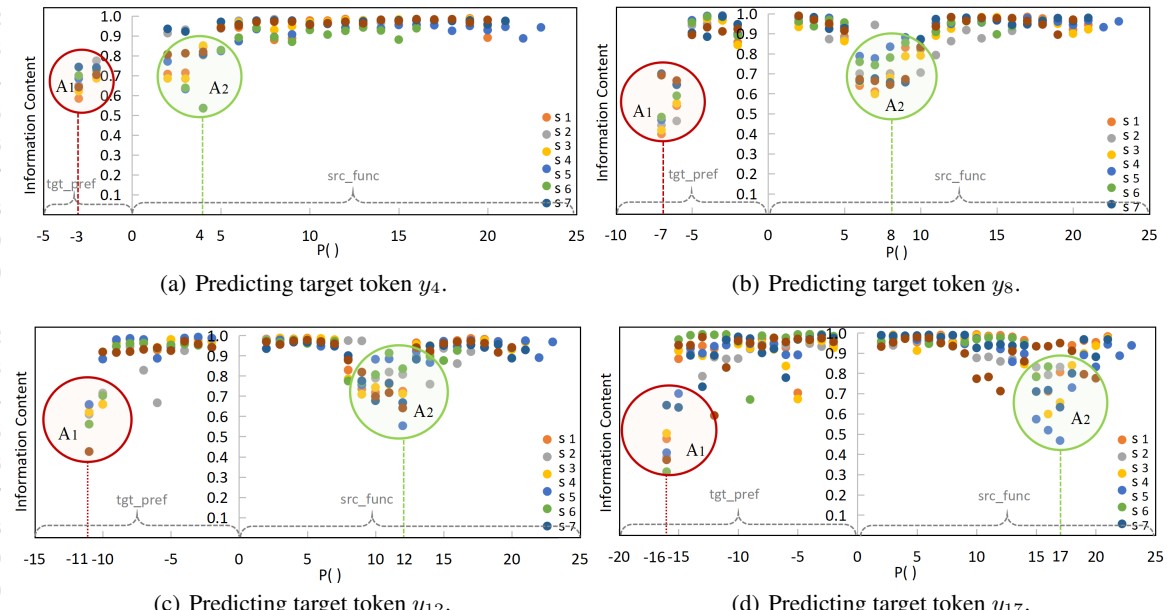

Figure 2: **Importance Distribution of High-Information Tokens in Small Samples.** We focus on target tokens $y_{j \in 4,8,12,17}$ at various positions to explore whether next-token predictions depend on tokens with specific positions. For each token $y_j$, we randomly select seven pre-target inputs $\langle$src_func, tgt_pref$\rangle$, denoted as $\{s_1, s_2, \ldots, s_7\}$. Then, we systematically corrupt individual tokens in sequence and employ the causal tracing method to evaluate the information content of each token. Here, $A_1$ and $A_2$ represent the clustered regions of high-information tokens, and the $x$-axis represents the positions of the tokens.

## 3.1 DEFINITIONS AND NOTATION

In the code correction scenario, a potentially incorrect source code $x$ (referred to as src_func) is mapped to a corrected code $y$ through LLM. Both $x$ and $y$ are functions in the considered programming language like Python, and each is represented by a sequence of tokens $x = [x_1, x_2, \ldots, x_M]$ and $y = [y_1, y_2, \ldots, y_N]$.

Consider an autoregressive Transformer language model (Irie et al., 2019), where all previously generated tokens $y_{1:j-1}$ are treated as additional input when generating the next token $y_{j \in [1,N]}$ (Vaswani et al., 2017; Dou & Gales, 2022; Goodman et al., 2020). We designate $y_j$ as a *target token* and $y_{1:j-1}$ as tgt_pref (i.e. a target prefix). For a target token $y_j$, since the output distribution $p(y_j|x, y_{1:j-1})$ is conditioned on both the src_func $x$ and the tgt_pref $y_{1:j-1}$, we define the pre-target input as $\langle$src_func, tgt_pref$\rangle$ or $c_j = (x, y_{1:j-1})$.

Inspired by Khandelwal et al. (2020), we use the mapping $f$ from the pre-target input to an intermediate representation of the Transformer decoder (i.e. the output of the final layer of a Transformer before the linear layer, see (Vaswani et al., 2017)) to obtain a vector representation of $\langle$src_func, tgt_pref$\rangle$. For the pre-target input $c_j$, this mapping yields a internal representation $k_j = f(x, y_{1:j-1})$.

*Definition 1*: Given two tokens $x_i$, $y_j$ across different contexts, if both exhibit analogous functionality and purpose, we define $x_i$ to be the *counterpart*[2] of $y_j$.

*Definition 2*: Given the target token $y_j$, we define its counterpart $x_i$ in $src\_func$ and the preceding token $y_{j-1}$ as the *core tokens* of $\langle$src_func, tgt_pref$\rangle$, denoted as $x_i^*$ and $y_{j-1}^*$, respectively.

*Definition 3*: Consider a token in $\langle$src_func, tgt_pref$\rangle$, we define the function $p()$ to capture the position of the token. Specifically, if the token belongs to src_func, then $p(x_i) = i$; and if the token belongs to tgt_pref, then $p(y_j) = -j$.

---

[2]In this paper, we consider only cases where a counterpart of the target token exists in the src_func.

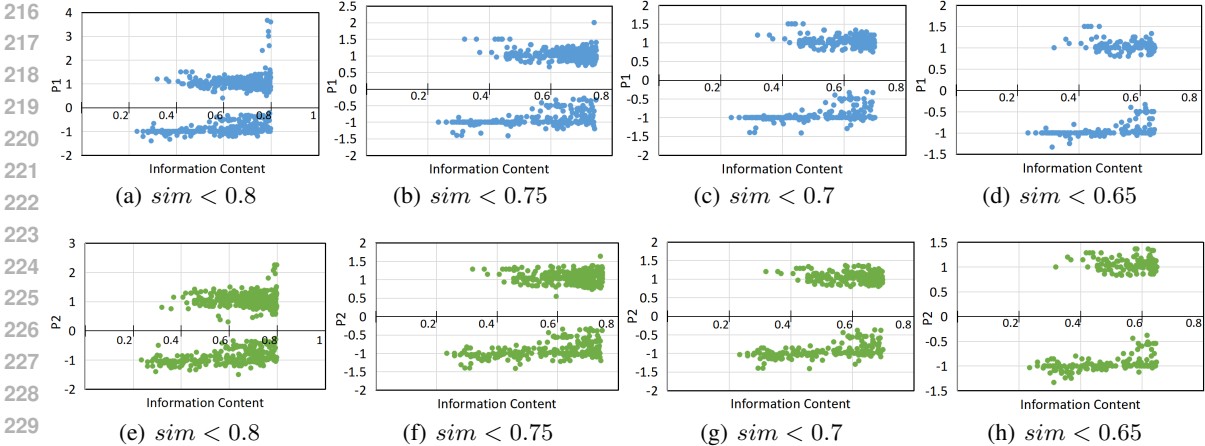

Figure 3: **Cluster Analysis of High-Information Tokens in $\langle$src_func, tgt_pref$\rangle$.** We utilize two normalization methods to examine whether high-information tokens exhibit clustering in large samples. Building upon this, we further investigate the importance distribution pattern of these high-information tokens with varying levels of information content.

## 3.2 CAUSAL TRACING METHOD

Large language models can be viewed as intermediaries that transform low-dimensional text sequences into high-dimensional internal representations. In this section, we introduce a *causal tracing method* designed to quantify the information content of individual tokens. This method involves perturbing individual tokens in $\langle$src_func, tgt_pref$\rangle$ and employs the *dot product similarity* to analyze fluctuations in the internal representation.

- **Clean representation:** Given a pre-target input $c_j$, we utilize the Transformer model to convert this clean pre-target input into an internal representation, denoted as $k_j$.

- **Corrupted representation:** We corrupt an individual token in $\langle$src_func, tgt_pref$\rangle$, resulting in a corrupted pre-target input $c_j'$. We then apply the Transformer model to convert this corrupted pre-target input $c_j'$ into an internal representation, denoted as $k_j'$.

- To assess the contribution of the corrupted token to the internal representation, we employ dot product similarity $sim()$ to measure the difference between $k_j$ and $k_j'$. This method provides a metric for quantifying the information content of the token (Luo et al., 2018). A lower similarity score indicates higher information content for the token, suggesting that the corrupted token has lost more critical information, as the perturbation significantly distorts the internal representation. In contrast, a higher score reflects lower information content, implying a smaller impact on the internal representation.

$$sim(c_j, c_j') = \frac{k_j \cdot k_j'}{\parallel k_j \parallel \parallel k_j' \parallel}$$

We performed single-token perturbations on 3,650 samples and calculated the similarity between these corrupted representations and clean representations. As illustrated in Figure 4, we observe a distinct clustering phenomenon, with the majority of samples exhibiting similarity scores between 0.8 and 1, while only a small fraction displays scores below 0.8. Based on this, we refer to a token with a similarity score below 0.8 as a *high-information token*, while a token with a score above 0.8 is referred to as a *low-information token*.

## 3.3 MULTI-PHASE CAUSAL TRACING PROCESS

Based on the causal tracing method, we propose a multi-phase causal tracing process to analyze the importance distribution pattern of high-information tokens. This process consists of three phases: **A)** Observation of the distribution pattern of high-information tokens in small samples; **B)** Assess-

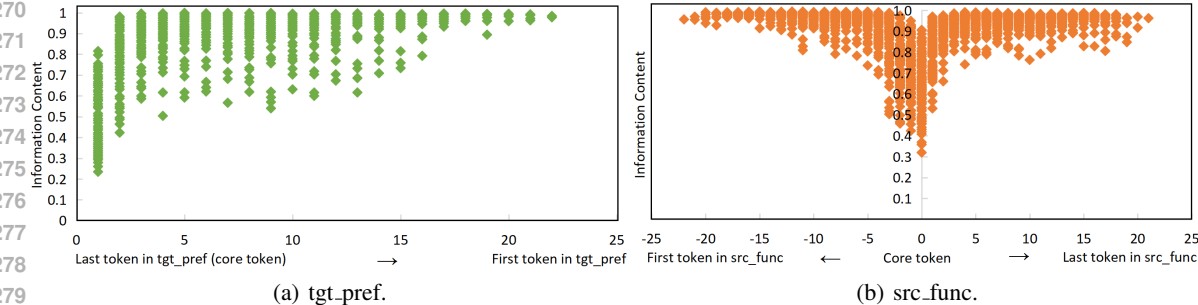

(a) tgt_pref.          (b) src_func.

Figure 4: **High-Information Tokens Cluster Around Core Tokens.** We performed an extensive analysis on 3,650 samples, utilizing two core tokens from src_func and tgt_pref as reference points. We found that high-information tokens tend to cluster around the $x = 0$ axis (i.e., core token), with their density gradually decreasing as the distance from this axis increases.

ment of whether high-information tokens exhibit clustering phenomena in large samples; and **C)** Evaluation of cluster centers for high-information tokens in large samples.

**A) High-Information Tokens Cluster Around Core Tokens in Small Samples.** In this section, we employ the causal tracing method to evaluate the contribution of each token, observing the importance distribution of high- and low-information tokens. Initially, we focus on target tokens $y_{j \in 4,8,12,17}$ at various positions, aiming to investigate whether next-token predictions depend on tokens from specific positions in the pre-target input. Then, for each target token $y_j$, we randomly select seven pre-target inputs $\langle \text{src\_func}, \text{tgt\_pref} \rangle$. We systematically corrupt individual tokens in sequence and apply the causal tracing method to evaluate the information content of each token, analyzing the importance distribution of high- and low-information tokens.

As shown in Figure 2, we observe a distinct clustering phenomenon in the importance distribution of high-information tokens. For the target tokens $y_{j \in 4,8,12,17}$, high-information tokens tend to cluster around the $x = j$ and $x = -(j - 1)$ axes. By examining the source documents, we find that these clustered points typically correspond to core tokens, and the associated code snippets often exhibit functional equivalence.

**B) Clustering Phenomena of High-Information Tokens in Large Samples.** As illustrated in Figure 2, we observed clustering of high-information tokens in both src_func and tgt_pref. In this section, we will further explore the existence of clustering phenomena in a large dataset. Specifically, we systematically corrupt individual tokens in $\langle \text{src\_func}, \text{tgt\_pref} \rangle$ and then employ a causal tracing method to assess the information content of each token across 3,650 samples. Following this, we identify high-information tokens with similarity scores below the thresholds of $\{0.8, 0.75, 0.7, 0.65\}$, denoting their positions as the set P. For example, if high-information tokens in a pre-target input are $\{x_3, x_4, x_5, x_6, y_4, y_5\}$, the position set P $= \{p(x_3), p(x_4), p(x_5), p(x_6), p(y_4), p(y_5)\}$. To analyze the importance distribution pattern of these tokens in $\langle \text{src\_func}, \text{tgt\_pref} \rangle$, we standardized their positions using the following formulas:

$$\text{P}_1 = \frac{\text{P}}{\text{median}(|\text{P}|)}, \quad \text{P}_2 = \frac{\text{P}}{\text{mean}(|\text{P}|)}$$

The standardized positions $\text{P}_1$ and $\text{P}_2$ facilitate a clearer observation of the distribution pattern by normalizing their positional data. It allows for more meaningful comparisons across different pre-target inputs and mitigates the effects of varying sequence lengths and token positions. As shown in Figure 3, high-information tokens with varying information content exhibit clear clustering behavior under both criteria, $\text{P}_1$ and $\text{P}_2$, with noticeable concentrations along the $y = 1$ and $y = -1$ axes. This non-random clustering phenomenon suggests that there may be intrinsic associations among high-information tokens.

**C) Clustering of High-Information Tokens Around Core Tokens in Large Samples.** Based on Part B, we identified a remarkable clustering phenomenon among high-information tokens, which consistently cluster around the $y = 1$ and $y = -1$ axes (as shown in Figure 3), indicating the existence of cluster centers. In Part A, we found that high-impact tokens cluster around core tokens

in small samples. To further investigate this relationship, we conducted an extensive analysis of 3,650 samples, utilizing two specific clustering centers, $p(x_i^*)$ and $p(y_{j-1}^*)$, as coordinate origins to examine the distribution characteristics of the surrounding tokens. The goal of this analysis was to examine the concentration of high-information tokens around the core tokens and to assess how this information density varies with distance.

In Figure 4, we observe that high-information tokens tend to cluster around the $x = 0$ axis (i.e., core tokens), with the density gradually decreasing as the distance increases. Specifically, in src_func, high-information tokens are densely packed within a radius of four tokens from the core token, while in tgt_pref, high-information tokens cluster within a radius of three tokens from the core token.

### 3.4 IMPACTFUL INFORMATION SOURCE HYPOTHESIS: IMPORTANT POSITION RULE

Building upon the multi-phase causal tracing process, we uncover a consistent pattern of the high-information tokens: Given a target token $y_j$, high-information tokens cluster within a 4-token radius around the core token $x_i^*$ in src_func and within a 2-token radius around the core token $y_{j-1}^*$ in tgt_pref. This pattern can be expressed as $(x_{max\{i-4,1\}:min\{i+4,M\}}^*, y_{max\{j-3,1\}:j-1}^*)$, known as the **Important Position Rule** (IPR).

*Hypothesis*: In LLM-based code inference, code snippets identified by the importance position rule are decisive for next-token prediction.

Based on the importance position rule, we define the code snippets extracted from the pre-target inputs $\langle$src_func, tgt_pref$\rangle$ as *IPR-based code snippets* (also referred to as *decisive code snippets*), denoted as $\langle$src_frag, tgt_frag$\rangle$. Specifically, src_frag= $x_{max\{i-4,1\}:min\{i+4,M\}}^*$ and tgt_frag= $y_{max\{j-3,1\}:j-1}^*$.

## 4 ANALYSIS OF IPR IN CODE INFERENCE

To validate that IPR-based code snippets are decisive for LLM-based code inference, we consider the following two questions: **Q1:** Can only code snippets identified by IPR reliably predict the target token? **Q2:** Does IPR exhibit strong generalization, making it applicable across diverse programming languages, tasks, and LLMs? (See Section 4.2 for the answer)

### 4.1 EXPERIMENTAL DETAILS

**Methods.** For **Q1**, we extract the IPR-based code snippet and use it to generate the next token. Specifically, given a target token $y_j$, we extract IPR-based code snippets $\langle$src_frag, tgt_frag$\rangle$ = $(x_{max\{i-4,1\}:min\{i+4,M\}}^*, y_{max\{j-3,1\}:j-1}^*)$ from pre-target inputs $\langle$src_func, tgt_pref$\rangle = (x, y_{j-1})$. Then, we use these short code snippets as inputs to the LLM, calculating the *success rate* by comparing the next generated token $y_j'$ with the original target token $y_j$.

$$\text{Success Rate} = \frac{\sum_{i=1}^N \mathbb{I}(y_i' = y_i)}{N}$$

Where, $N$ denotes the total number of samples. $y_i$ represents the original target token, which generated by $\langle$src_func, tgt_pref$\rangle$. $y_j'$ represents the predicted token generated by IPR-based code snippets $\langle$src_frag, tgt_frag$\rangle$. A match $y_j = y_j'$ is considered a success, indicating the significance of the IPR-based code snippet in next-token prediction. Conversely, a mismatch suggests the model relied on broader context rather than the IPR-based code snippet.

For **Q2**, based on the above method, we evaluate the generalizability of IPR across various models, tasks, and programming languages. Specifically, we assessed the next-token prediction performance of IPR-based code snippets across three distinct tasks: code correction, code translation, and code completion. These tasks utilize models such as CodeLlama-7b-Instruct, CodeLlama-13b-Instruct, CodeLlama-34b-Instruct, gpt-3.5-turbo[3], and gpt-4-turbo[4], covering programming languages including C++, Java, and Python (Roziere et al., 2023; Ouyang et al., 2022).

---

[3]https://platform.openai.com/docs/models#gpt-3-5-turbo
[4]https://platform.openai.com/docs/models#gpt-4-turbo-and-gpt-4

**Test Dataset.** Different models interpret and handle context in distinct ways during model inference, leading to variations in the inference results (i.e., tgt_pref) for the same src_func. Therefore, it is necessary to construct model-specific ⟨src_func, tgt_pref⟩ language pairs for each model. Based on this, we created specialized IPR-based snippet datasets for CodeLlama-7b-Instruct, CodeLlama-13b-Instruct, CodeLlama-34b-Instruct, gpt-3.5-turbo, and gpt-4-turbo, enabling the analysis of different models' inference performance. (See Appendix A.2 for details on test dataset generation.)

## 4.2 RESULTS AND DISCUSSION

**Evaluating IPR: Code Correction.** In this phase, we validated the role of the important position rule in code correction tasks across various large language models in the contexts of Java, C++, and Python. As shown in Table 1, models such as CodeLlama-7b-Instruct, CodeLlama-13b-Instruct, and CodeLlama-34b-Instruct demonstrated remarkable *success rates* in matching the original target token, attaining rates between 91.86% and 94.13% while utilizing only 12 high-information tokens. In comparison, the performance of gpt-3.5/4-turbo was relatively lower. This difference can be attributed to their design focus: both models are primarily optimized for natural language processing tasks, emphasizing the understanding and generation of natural language rather than reasoning and generation in programming languages. Compared to gpt-3.5-turbo, researchers enhanced gpt-4-turbo's capabilities in code generation tasks (OpenAI et al., 2024), which contributes to the higher success rate. These results indicate that in code correction, the next-token prediction largely relies on high-information tokens derived from the important position rule. Furthermore, the consistently high success rates of IPR-based code snippets across various programming languages and LLMs highlight the strong generalization capability of IPR.

Table 1: Performance of IPR-Based Code Snippets for Next-Token Prediction in Code Correction.

|  | CodeLlama-7b-Instruct | CodeLlama-13b-Instruct | CodeLlama-34b-Instruct | gpt-3.5-turbo | gpt-4-turbo | Ave_1 |
|---|---|---|---|---|---|---|
| C++ | 91.86% | 93.70 % | 92.60 % | 56.77 % | 69.69 % | 80.92 % |
| Java | 92.41 % | 94.13% | 93.00 % | 52.55 % | 72.50 % | 80.92 % |
| Python | 92.21 % | 92.79 % | 92.82 % | 54.05 % | 73.25 % | 81.02 % |
| Ave_2 | 92.16 % | 93.54 % | 92.81 % | 54.46 % | 71.81 % | - |

**Evaluating IPR: Code Translation.** We evaluate the contribution of IPR-based code snippets in the following translation tasks: C++ → Python, C++ → Java, Java → Python, Java → C++, Python → Java, and Python → C++. As illustrated in Table 2, we found that even using code snippets with only 12 tokens for next-token prediction, all models achieve significant success rates in matching the original target token. Specifically, CodeLlama-7b-Instruct, CodeLlama-13b-Instruct, CodeLlama-34b-Instruct, and gpt-4-turbo attain success rates ranging from 80.98% to 70.11%. Furthermore, we observed that all models performed particularly well in the C++ → Java and Java → C++ tasks. This is because Java is developed based on C++, retaining much of its syntax and core programming paradigms. In contrast, despite the significant differences in structure, keywords, and syntax between Python and C++/Java, specialized code LLMs such as CodeLlama-7b-Instruct, CodeLlama-13b-Instruct, and CodeLlama-34b-Instruct still achieved success rates ranging from 64.94% to 80.00%. These results indicate that LLM inference in code translation tasks heavily relies on IPR-based code snippets, further highlighting the strong generalization capability of IPR in code translation tasks.

Table 2: Performance of IPR-Based Code Snippets for Next-Token Prediction in Code Translation.

|  | CodeLlama-7b-Instruct | CodeLlama-13b-Instruct | CodeLlama-34b-Instruct | gpt-3.5-turbo | gpt-4-turbo | Ave_1 |
|---|---|---|---|---|---|---|
| C++ → Java | 86.43 % | 92.19 % | 92.38 % | 55.33 % | 78.56 % | 80.98 % |
| C++ → Python | 74.01 % | 80.00% | 73.52 % | 49.15 % | 76.03 % | 70.54 % |
| Java → C++ | 91.41 % | 94.12 % | 92.73 % | 60.27 % | 83.21 % | 84.35 % |
| Java → Python | 70.52 % | 76.54 % | 73.56 % | 39.80 % | 74.70 % | 67.02 % |
| Python → C++ | 71.31 % | 72.95 % | 64.94% | 39.26 % | 58.39 % | 61.37 % |
| Python → Java | 67.09 % | 70.10 % | 66.04 % | 32.46 % | 49.75 % | 57.09 % |
| Ave_2 | 76.80 % | 80.98% | 77.20 % | 46.05 % | 70.11% | - |

**Evaluating IPR: Code Completion.** In this section, we tested the role of IPR in code completion tasks. Here, we will consider only the impact of tgt_frag on the inference process. As illustrated in

Table 3: Performance of IPR-Based Code Snippets for Next-Token Prediction in Code Completion.

|  | CodeLlama-7b-Instruct | CodeLlama-13b-Instruct | CodeLlama-34b-Instruct | gpt-3.5-turbo | gpt-4-turbo |
|---|---|---|---|---|---|
| C++_tok3 | 27.12% | 28.45% | 26.81% | 27.49% | 22.63% |
| C++_tok8 | 63.21% | 63.35% | 62.55% | 40.85% | 43.64% |
| Java_tok3 | 34.00% | 27.89% | 29.16% | 22.38% | 28.46% |
| Java_tok8 | 66.84% | 61.94% | 62.61% | 38.65% | 44.24% |
| Python_tok3 | 28.03% | 28.15% | 30.24% | 21.49% | 30.95% |
| Python_tok8 | 50.27% | 50.78% | 51.32% | 37.93% | 44.74% |
| Ave_tok3 | 29.72% | 28.16% | 28.74% | 23.79% | 27.35% |
| Ave_tok8 | 60.11% | 58.69% | 58.82% | 39.14% | 44.21% |

Figure 4, we observe a clear trend: the closer a token is to the core token, the greater its information content, which plays a more significant role in guiding the model's inference. Building on this observation, we evaluated the inference performance using two different configurations of the tgt_frag: i) a 3-token snippet $\{y_{j-3}, y_{j-2}, y_{j-1}\}$, which focuses on a few tokens preceding the target token, and ii) an 8-token snippet $\{y_{j-8}, ..., y_{j-1}\}$, which includes a broader context from the prefix. As shown in Table 3, with only 3 tokens, the success rate ranges from 23.79% to 29.72%, indicating that even a limited number of high-information tokens can still contribute to code inference. When the context is expanded to 8 tokens, the success rate increases significantly. For the CodeLlama-7b-Instruct, CodeLlama-13b-Instruct, and CodeLlama-34b-Instruct models, the success rate rises to between 58.69% and 60.11%, while the gpt-3.5-turbo and gpt-4-turbo models exhibit lower but still notable success rates, ranging from 39.14% to 44.21%. These results highlight the key role of IPR-based code snippets in LLM inference, demonstrating their remarkable generalization capabilities across multi-language and multi-model scenarios, thereby providing meaningful insights into next-token prediction.

## 5 APPLICATION: IPR-BASED KNOWLEDGE EDITING

In this section, we introduce an application of IPR: knowledge editing for LLMs in the context of programming languages. We integrate IPR with the ROME approach (Meng et al., 2023), effectively correcting errors in Java to Python translations by updating middle layer weights.

### 5.1 IPR-BASED ROME APPROACH

Existing knowledge editing techniques are constrained by the inherent ⟨subject, relation, object⟩ structure of natural language. In this paper, we elucidate the crucial role of IPR in next-token prediction, where IPR-based code snippets, src_frag and tgt_frag, exhibit significant semantic and syntactic correlations. Furthermore, the core token in src_frag exhibits analogous functionality and purpose to the target token, similar to the relationship between "subject" and "object". This finding indicates the potential of IPR to support knowledge editing in the context of programming languages.

Building on this, we integrate ROME with IPR to perform knowledge editing on the mid-layer feed-forward module of the CodeLlama-7b-Instruct model, thereby correcting errors in Java to Python translation. Specifically, for a failed Java-Python translation pair, we consider the error as the target token and employ the IPR to extract the corresponding code snippets ⟨src_frag, tgt_frag⟩ as the basis for knowledge editing. For the tuple ⟨src_frag, tgt_frag, corrected_error⟩, we treat the core token of src_frag as the "subject" and the corrected_error as the "new object", thereby generating a new key-value pair $(k^*, v^*)$. This key-value pair allows us to update the weight matrix using the equation $Wk^* = v^*$, where $k^*$ and $v^*$ are defined as follows:

$$k^* = \frac{1}{N} \sum_{j=1}^{N} \sigma(W_{fc}^{(l^*)} \cdot \gamma(a_{[x],i}^{(l^*)} + h_{[x],i}^{(l^*-1)})),$$

$$v^* = \arg\min_{z} \left( \frac{1}{N} \sum_{j=1}^{N} -\log P_{G(m_i^{l^*}:=z)}[o^*|x_j + p] + D_{KL}(P_{G(m_i^{l^*}:=z)}[x|p'] \| P_G[x|p']) \right)$$

where $N$ represents the sample size, $\sigma$ denotes the activation function, $W^{(l^*)}_{fc}$ and $W^{(l^*)}_{prof}$ refer to the weight matrices of the fully connected layer at layer $l^*$, $\gamma$ is the feature extraction function, $a_{[x],i}^{(l^*)}$ indicates the activation value of input $x$ at layer $l^*$, $h_{[x],i}^{(l^*-1)}$ represents the activation

value of input $x$ at the previous layer. $-\log P_{G(m_{i'}^{l*}:=z)}[o^*|x_j + p]$ aims to find a vector $z$ that, when substituted as the output of the MLP at the $i$-th token in the subject part, enables the network to predict the target object $o^*$ given the factual prompt $p$. $D_{KL}(P_{G(m_{i'}^{l*}:=z)}[x|p']\|P_G[x|p'])$ minimizes the KL divergence between the predictions for the prompt $p'$ and the original model's predictions.

## 5.2 RESULTS AND DISCUSSION

**Details.** First, we utilized a dataset of 25,227 Java functions as input to CodeLlama-7b-Instruct and collected 2,975 incorrect Python translations through unit testing. We then categorized these errors based on specific tokens and selected two main errors: indexOf (Java) $\rightarrow$ index (Python), accounting for 5.41%; and equals (Java) $\rightarrow$ equals (Python), accounting for 2.72%. Based on this, we created specialized Java datasets targeting these two error types to evaluate the effectiveness of knowledge editing. The main reason for selecting these errors is that their corresponding correct translations are both single tokens, allowing for correction through a one-time knowledge edit, which is applicable to the ROME method. Next, we consider performing knowledge editing on the $19^{th}$ MLP layer of CodeLlama-7b-Instruct, primarily for two reasons: (1) Meng et al. (2023) proposed that the mid-layer feed-forward module plays an important role in storing factual associations, and (2) our analysis using Logit Lens (nostalgebraist, 2020) indicated that the $19^{th}$ MLP layer has a greater impact on the final output (see Appendix Figure 7).

Table 4: Performance of Knowledge Editing for Correcting Translation Errors (Java $\rightarrow$ Python).

| | Generation of $(k^*, v^*)$ | Succ | Total | Succ Rate |
|---|---|---|---|---|
| $[A]_1$ | $\langle$src_frag, tgt_frag$\rangle$: idx = uri . **[** indexOf **]** ( " : " python = uri . corrected_error: find | 101 | 161 | **62.73%** |
| $[B]_1$ | $\langle$src_frag, tgt_frag$\rangle$: idx = uri . indexOf ( " **[** : **]** " python = uri . corrected_error: find | 42 | 161 | 26.09% |
| $[C]_1$ | $\langle$src_frag, tgt_frag$\rangle$: idx **[** = **]** uri . indexOf ( " : " python = uri . corrected_error: find | 45 | 161 | 27.95% |
| $[A]_2$ | $\langle$src_frag, tgt_frag$\rangle$: null : s1 . **[** equals **]** ( s2 ) ; python : return s1 corrected_error: == | 61 | 81 | **75.31%** |
| $[B]_2$ | $\langle$src_frag, tgt_frag$\rangle$: null : s1 . equals ( s2 **[** ) **]** ; python : return s1 corrected_error: == | 20 | 81 | 24.69% |
| $[C]_2$ | $\langle$src_frag, tgt_frag$\rangle$: **[** null **]** : s1 . equals ( s2 ) ; python : return s1 corrected_error: == | 25 | 81 | 30.86% |

* $[ \cdot ]_1$ indicates the model editing performed on the error: indexOf (java) $\rightarrow$ index (python). Here, the error "index" should be corrected to "find".
* $[ \cdot ]_2$ indicates the model editing performed on the error: equals (java) $\rightarrow$ equals (python). Here, the error "equals" should be corrected to "==".
* $[ \cdot ]$ indicates the editing position.

**Evaluation.** For these two types of errors, we performed model editing and assessed the correction performance through unit testing. As shown in Table 4, we select core tokens in src_func as editing points. To validate the effectiveness of core tokens, we also included two additional comparison positions: the core token in src_func [A], the token following the core token [B], and the token preceding the core token [C]. In Table 4, the results indicate that editing at position [A] significantly enhanced the correction rates for both errors, achieving rates of 62.73% and 75.31%, respectively. In contrast, the correction rates for edits at positions [B] and [C] were limited, ranging only from 24.69% to 30.86%. We speculate that position [A] functions similarly to the subject in syntactic structures, demonstrating a strong association with the object and profoundly influencing the flow and guidance of information. This suggests that the important position rule can provide valuable support for knowledge editing in the context of programming languages.

## 6 CONCLUSION

In this paper, we introduce a causal tracing method to identify high-information tokens, enabling a precise understanding of how Transformer models extract information from pre-target inputs in the context of programming languages. Based on this, we reveal a consistent pattern of high-information tokens, named the important position rule. Then, we thoroughly evaluated the performance of IPR across various models, tasks, and programming languages. Furthermore, we successfully applied IPR to knowledge editing, proving that IPR can provide support for knowledge editing and interpretable inference.

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

# A  APPENDIX

## A.1  DATASET

We first downloaded raw source code datasets for Java, Python, and C++ from Google BigQuery[5]. Referring to the work of Roziere et al. (2020)[6], we then performed data preprocessing and obtained a dataset consisting of 11.98 million Java functions, 8.83 million Python functions, and 6.54 million C++ functions. All subsequent datasets used in our study were generated from this initial dataset.

### A.1.1  IPR-BASED SNIPPET DATASETS

Different models interpret context and preferences in distinct ways, leading to varying tgt_pref outputs for the same src_func. Therefore, it is necessary to construct dedicated IPR-based snippet datasets for each model. Based on this, we respectively construct specialized IPR-based snippet dataset for CodeLlama-7b-Instruct, CodeLlama-13b-Instruct, CodeLlama-34b-Instruct, gpt-3.5-turbo, and gpt-4-turbo, targeting tasks such as code translation, code correction, and code completion.

**Code Translation:** We randomly sample subdatasets of Java, Python, and C++ from the initial dataset. These subdatasets serve as input sequence datasets, representing the src_func in the pre-target inputs. (In Figure 5, we show the length distribution of the input sequences in the dataset.)

- *Performing Code Translation on Input Sequence Datasets*: First, we feed the input sequence datasets for C++, Java, and Python into the models, including CodeLlama-7b/13b/34b-Instruct and gpt-3.5/4-turbo. We then translate each input into the other two target languages (e.g., translating the C++ into Python and Java), collecting pairs of (input sequence, output sequence), denoted as $(x, y)$. In this step, we follow the natural output of each model without requiring verification of the output's correctness.

- *Extracting Pre-Target Inputs and Target Tokens*: For each $(x, y)$, we set tgt_pref as the first half of the output sequence. Based on this, we extract a collection of target tokens and their corresponding pre-target inputs, where pre-target input $\langle \text{src\_func}, \text{tgt\_pref} \rangle = (x, y_{1:\text{len}(y)/2})$, and the target token is $y_{\text{len}(y)/2+1}$.

- *IPR-Based Snippet Identification*: Finally, we collect target tokens $y_{\text{len}(y)/2+1}$ along with the IPR-based code snippets $\langle \text{src\_frag}, \text{tgt\_frag} \rangle$ extracted from pre-target inputs $(x, y_{1:\text{len}(y)/2})$, and then perform a deduplication process. This approach allows us to construct specialized IPR-based snippet datasets, each tailored to a specific model, such as CodeLlama-7b-Instruct, CodeLlama-13b-Instruct, CodeLlama-34b-Instruct, gpt-3.5-turbo, and gpt-4-turbo.

**Code Correction:** We randomly extract subdatasets of Java, Python, and C++ from the initial dataset. Using these subdatasets, we construct input sequence datasets for the code correction task.

- *Building input sequence datasets*: In existing literature and within the knowledge base of LLMs Thomas & Hunt (2019); Sedgewick & Wayne (2017); Agans (2002), there is a comprehensive and extensive collection of common programming errors across different languages. For the above Java, Python, and C++ subdatasets, we leverage the LLM to simulate realistic coding mistakes, injecting $2 \sim 3$ typical errors into each sample, thereby generating error Java/Python/C++ datasets. These datasets correspond to the input sequences src_func in the pre-target inputs. This process involves deliberately introducing common syntactical, logical, or runtime errors specific to each language, such as incorrect variable assignments, misplaced parentheses, or improper function calls. (Figure 5 shows the length distribution of input sequences in the dataset.)

- *Performing Code Correction on Input Sequence Datasets*: Next, we feed the input sequence datasets into models such as CodeLlama-7b/13b/34b-Instruct and gpt-3.5/4-turbo for code correction. For each model, we collect pairs of (input sequence, output sequence), denoted as $(x, y)$.

---

[5]https://cloud.google.com/blog/products/gcp/github-on-bigquery-analyze-all-the-open-source-code
[6]https://github.com/facebookresearch/TransCoder

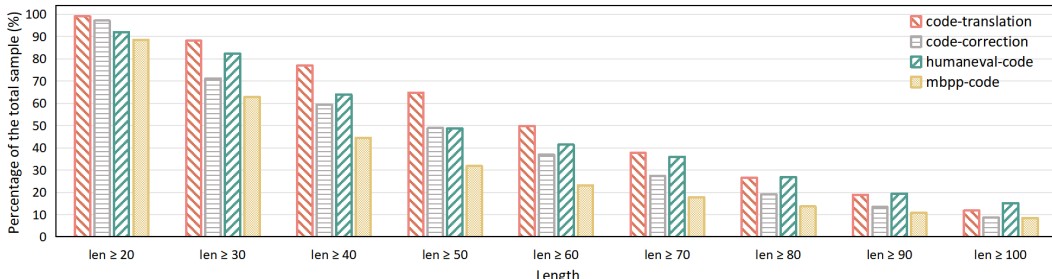

Figure 5: **Length Distribution of Input Sequence Datasets in Code Translation and Code Correction Tasks.** For both the code translation and code correction tasks, we conducted a statistical analysis of the input sequence lengths across all models, including CodeLlama-7b/13b/34b-Instruct and gpt-3.5/4-turbo, in the contexts of Java, Python, and C++. To provide a meaningful comparison, we also examined the length distributions of widely used datasets, such as Humaneval and MBPP. Our analysis reveals that the length distributions of the datasets used for code translation and code correction tasks are closely aligned. Here, the sequence length refers to the number of tokens. The $x$-axis represents $len \geq n$, indicating that the sequence length is greater than or equal to $n$, while the $y$-axis represents the proportion of samples with $len \geq n$. (See Table 5 for details.)

- *Extracting Pre-Target Inputs and Target Tokens*: Similar to the code translation task, for each language pair $(x, y)$, we define tgt_pref $= y_{1:\text{len}(y)/2}$ and the target token $= y_{\text{len}(y)/2+1}$. For each LLM, we generate specialized datasets consisting of pre-target inputs $\langle \text{src\_func}, \text{tgt\_pref} \rangle = (x, y_{1:\text{len}(y)/2})$ and their corresponding target tokens $y_{\text{len}(y)/2+1}$.

- *IPR-Based Snippet Identification*: Finally, we extract the target tokens $y_{\text{len}(y)/2+1}$ and the IPR-based code snippets $\langle \text{src\_frag}, \text{tgt\_frag} \rangle$ from the pre-target inputs $(x, y_{1:\text{len}(y)/2})$, followed by a deduplication process. This process enables us to build IPR-based fragment datasets tailored for each model, including CodeLlama-7b-Instruct, CodeLlama-13b-Instruct, CodeLlama-34b-Instruct, gpt-3.5-turbo, and gpt-4-turbo.

**Code Completion** We randomly select Java, Python, and C++ subdatasets from the initial dataset. Based on this, we construct input sequence datasets for the code completion task.

- *Building input sequence datasets*: We extract the first 20 tokens from the Java, Python, and C++ subdatasets to construct the input sequence datasets for further processing.

- *Performing Code Completion on Input Sequence Datasets*: Subsequently, we feed the Java, Python, and C++ input sequence datasets into models, including CodeLlama-7b/13b/34b-Instruct and gpt-3.5/4-turbo, for code completion. For each model, we collect pairs of input and output sequences, denoted as $(x, y)$, where the output sequence $y$ represents the code continuation based on the input sequence $x$.

- *Extracting Pre-Target Inputs and Target Tokens*: For each $(x, y)$ pair, we extract the first 20 tokens from the generated output sequence as *tgt_pref*, where the target token is $y_{21}$, and *tgt_pref* $= y_{1:20}$. Using this method, we construct specialized datasets for each LLM, consisting of the target token $y_{21}$ and its associated pre-target input, $\langle \text{src\_func}, \text{tgt\_pref} \rangle = (x, y_{1:20})$.

- *IPR-Based Snippet Identification*: In the code completion task, there is no direct functional equivalence between the output sequence $y$ and the input sequence $x$. Therefore, when identifying high-information code snippets for code completion, we exclude the input sequence src_func and focus solely on extracting high-information snippets from the target prefix tgt_pref. Based on IPR, we collect a dataset consisting of IPR-based snippets $\{y_{18}, y_{19}, y_{20}\}$ and the corresponding target tokens $y_{21}$. Additionally, we create a broader context dataset by extracting code snippets $\{y_{13}, y_{14}, ..., y_{20}\}$ and target tokens $y_{21}$ for further exploration. In this way, we build two code snippet datasets for CodeLlama-7b/13b/34b-Instruct and gpt-3.5/4-turbo, respectively.

Table 5: Length Statistics of Input Sequence Datasets in Code Translation and Correction Tasks.

| Length | Code Translation | Code Correction | HumanEval | MBPP |
|---|---|---|---|---|
| $len \geq 20$ | 99.09% | 97.19% | 92.07% | 88.50% |
| $len \geq 30$ | 88.14% | 71.28% | 82.32% | 62.83% |
| $len \geq 40$ | 77.05% | 59.48% | 64.02% | 44.56% |
| $len \geq 50$ | 64.86% | 48.93% | 48.78% | 31.83% |
| $len \geq 60$ | 49.90% | 36.90% | 41.46% | 23.20% |
| $len \geq 70$ | 37.77% | 27.38% | 35.98% | 17.76% |
| $len \geq 80$ | 26.69% | 19.11% | 26.83% | 13.66% |
| $len \geq 90$ | 18.98% | 13.47% | 19.51% | 10.88% |
| $len \geq 100$ | 11.82% | 8.79% | 15.24% | 8.52% |

### A.1.2 GENERATION OF 3,650 CORRUPTED SAMPLES

In the code correction task described in Section A.1.1, we collected specialized datasets consisting of pre-target inputs $\langle \text{src\_func}, \text{tgt\_pref} \rangle = (x, y_{1:\text{len}(y)/2})$ and their corresponding target tokens $y_{\text{len}(y)/2+1}$. Based on this, we first randomly selected one hundred Python pre-target inputs from CodeLlama-7b-Instruct. Subsequently, using the causal tracing method, we individually corrupted each token in the pre-target inputs, generating 3,650 corrupted samples. These samples were then used to quantify the importance of each token in the pre-target inputs, allowing us to visualize the importance distribution of high-information tokens.

### A.1.3 JAVA-PYTHON TRANSLATION DATASET

**Necessity of Building a Large-Scale Java-Python Translation Dataset:** In Section 5, we aim to collect common errors made by CodeLlama-7b-Instruct when translating Java to Python. While CodeLlama-7b-Instruct achieves a translation accuracy of 88.21% for Java $\rightarrow$ Python, commonly used datasets offer limited samples, with only 164 examples in HumanEval (Chen et al., 2021) and 974 examples in MBPP (Austin et al., 2021). Relying solely on these two datasets to identify common errors is unreliable, as their limited sample size does not support a comprehensive analysis of main errors, which may lead to biased results.

**Construction Process of the Java-Python Translation Dataset:** Following the work of Roziere et al. (2022), we built a Java-Python translation dataset[7]. First, we download the Java source code from Google BigQuery and apply the TransCoder-ST preprocessing pipeline for dataset filtering. Next, we set a maximum runtime of 20 seconds for each process and use EvoSuite to generate Java test cases. Here, the unit test cases are created based on two criteria: a mutation score above 0.9 and at least two assertions. Through this process, we collect 82,665 Java functions with corresponding unit test cases. Subsequently, we feed the processed Java functions, along with their corresponding test cases, into TransCoder-ST (Java $\rightarrow$ Python, beam_size = 1).

For each Java function, we generate a corresponding Python function along with its associated unit test cases. Then, we execute the Python unit tests to validate the correctness of the translation. If the Python function passes all tests, we pair the Java function with the Python function to form a translation language pair. This process ensures the accuracy of both the Python functions and their corresponding unit tests. After removing duplicate entries from the processed dataset, we obtain a final Java-Python translation dataset comprising 25,227 language pairs, each accompanied by unit test cases. (See example in Table 10)

### A.2 IMPACT OF HIGH-INFORMATION TOKEN COUNT ON NEXT-TOKEN PREDICTION

As shown in Figure 4, high-information tokens in src_func are primarily symmetrically distributed around the core token, with fewer high-information tokens as the distance increases. Similarly, in tgt_pref, high-information tokens are mainly concentrated around the core token, and their frequency decreases with increasing distance. This observation explains the rationale for extracting high-information code snippets using the format $(x^*_{max\{i-a,1\}:min\{i+a,M\}}, y^*_{max\{j-b,1\}:j-1})$.

---

[7]https://github.com/facebookresearch/CodeGen

We further explain the reasons for choosing $a = 4$ and $b = 3$ to define the important position rule. In code correction and translation tasks, we experimented with high-information code snippets of varying lengths, exploring different combinations of parameters, such as $(a, b) \in \{(6, 5), (5, 4), (6, 5), (4, 3), (3, 2), (2, 1)\}$, which are denoted as $IPR_{+2tok}$, $IPR_{+1tok}$, IPR, $IPR_{-1tok}$, and $IPR_{-2tok}$. Then, we use the $IPR+2tok$, $IPR+1tok$, IPR, $IPR-1tok$, and $IPR-2tok$ snippet datasets as inputs to the LLM, calculating the success rate by comparing the generated token $y_j'$ with the original target token $y_j$, respectively.

As shown in Tables 6 and 7, we systematically reduced the number of high-information tokens near the core tokens. We found that $IPR_{-2tok}$ performed poorly across all tests. Although $IPR_{-1tok}$ achieved good results in CodeLlama-7b/13b/34b-Instruct, its performance was suboptimal in gpt-3.5/4-turbo, particularly in gpt-3.5-turbo. When $a = 4$ and $b = 3$, we observed that IPR-based snippets consistently yielded strong results in relatively smaller code snippets, both in CodeLlama-7b/13b/34b-Instruct and gpt-3.5/4-turbo for code correction and translation tasks.

### A.3 ABLATION ANALYSIS —— IMPORTANCE OF IPR-BASED CODE SNIPPETS

To explore the role of IPR-based code snippets in next-token prediction, we consider the following two methods in code completion and code correction tasks: (1) removing the IPR-based code snippet from $\langle src\_func, tgt\_pref \rangle$, referred to as the $IPR_{remove}$ code snippet; (2) corrupting the IPR-based code snippet in $\langle src\_func, tgt\_pref \rangle$, referred to as the $IPR_{corrupt}$ code snippet. Then, we use the $IPR_{remove}$ / $IPR_{corrupt}$ snippet dataset as input to the LLM and compute the success rate by comparing the generated token $y_j'$ with the original target token $y_j$.

As shown in Tables 8 and 9, we find that when we remove or corrupt the IPR-based code snippets, the success rate consistently drops below 8%. Further, we considered whether it stemmed from the short length of $\langle src\_func, tgt\_pref \rangle$. Specifically, after removing 12 tokens (i.e., the length of the IPR-based code snippets), the remaining valid tokens in $\langle src\_func, tgt\_pref \rangle$ were too few, leading to a significant impact on next-token prediction. Based on this, we counted the length of src_func in both the code translation and code correction tasks in Table 5. We found that in the code translation task, 88.14% of src_func samples exceed 30 tokens, while in the code correction task, 71.28% of src_func samples exceed 30 tokens. It means that, despite the fact that the lengths of the $IPR_{remove}$ and $IPR_{corrupt}$ code snippets are significantly greater than that of $\langle src\_frag, tgt\_frag \rangle$, the success rate of next-token prediction remains notably low. This result suggests that IPR-based code snippets play a crucial role in next-token prediction during code inference.

### A.4 EXTRACTION OF NEXT GENERATED TOKEN

In experiments evaluating the effect of IPR on next-token prediction, it is essential to test its performance on CodeLlama-7b/13b/34b-Instruct models and gpt-3.5/4-turbo. For CodeLlama-7b/13b/34b-Instruct, we can directly extract the next generated token from the model's output. However, since gpt-3.5/4-turbo is accessible only via the API, we leverage prompt engineering by including the instruction, "*Please generate the next java token.*" in the prompt to guide the model to directly generate the next token. (Some examples are shown in Figures **??** and 6.)

Table 6: The Impact of the Number of High-Information Tokens on Next-Token Prediction in Code Translation Tasks.

| Task | Type | CodeLlama-7b-Instruct | CodeLlama-13b-Instruct | CodeLlama-34b-Instruct | gpt-3.5-turbo | gpt-4-turbo | Ave_1 |
|------|------|------------------------|-------------------------|-------------------------|----------------|--------------|--------|
| C++ → Java | *baseline* | 6.27% | 7.73 % | 7.36 % | 2.61 % | 7.53 % | 6.30 % |
| | $IPR_{-2tok}$ | 71.33 % | 73.14 % | 73.00 % | 23.43 % | 59.18 % | 60.02 % |
| | $IPR_{-1tok}$ | 86.82 % | 89.55 % | 91.95 % | 33.28 % | 74.16 % | 75.15 % |
| | IPR | 86.43 % | 92.19 % | 92.38 % | 55.33 % | 78.56 % | 80.98 % |
| | $IPR_{+1tok}$ | 89.53 % | 93.25 % | 91.45 % | 58.82 % | 82.38 % | 83.09 % |
| | $IPR_{+2tok}$ | 88.07 % | 93.85 % | 92.80 % | 64.05 % | 81.64 % | 84.08 % |
| C++ → Python | *baseline* | 11.54% | 10.19 % | 10.56 % | 5.48 % | 8.96 % | 9.35 % |
| | $IPR_{-2tok}$ | 52.35 % | 52.80 % | 49.69 % | 14.33 % | 60.58 % | 45.95 % |
| | $IPR_{-1tok}$ | 75.24 % | 73.40 % | 65.08 % | 32.45 % | 73.54 % | 63.94 % |
| | IPR | 74.01 % | 80.00 % | 73.52 % | 49.15 % | 76.03 % | 70.54 % |
| | $IPR_{+1tok}$ | 80.19 % | 82.39 % | 76.13 % | 47.73 % | 82.08 % | 73.70 % |
| | $IPR_{+2tok}$ | 83.40 % | 85.12 % | 81.73 % | 49.01 % | 83.50 % | 76.55 % |
| Java → C++ | *baseline* | 7.71% | 7.40 % | 7.00 % | 5.37 % | 8.60 % | 7.22% |
| | $IPR_{-2tok}$ | 76.22 % | 75.82 % | 76.01 % | 28.25 % | 66.42 % | 64.54 % |
| | $IPR_{-1tok}$ | 90.07 % | 92.37 % | 91.63 % | 44.31 % | 81.32 % | 79.94 % |
| | IPR | 91.41 % | 94.12 % | 92.73 % | 60.27 % | 83.21 % | 84.35 % |
| | $IPR_{+1tok}$ | 91.57 % | 94.77 % | 94.01 % | 65.05 % | 87.30 % | 86.54 % |
| | $IPR_{+2tok}$ | 91.63 % | 94.97 % | 93.81 % | 65.99 % | 86.62 % | 86.60 % |
| Java → Python | *baseline* | 10.65% | 11.35 % | 9.85 % | 5.00 % | 10.45 % | 9.46 % |
| | $IPR_{-2tok}$ | 50.24 % | 48.22 % | 49.77 % | 9.73 % | 58.33 % | 43.26 % |
| | $IPR_{-1tok}$ | 67.81 % | 71.69 % | 67.46 % | 18.37 % | 65.48 % | 58.16 % |
| | IPR | 70.52 % | 76.54 % | 73.56 % | 39.80 % | 74.70 % | 67.02 % |
| | $IPR_{+1tok}$ | 74.83 % | 80.78 % | 79.41 % | 33.11 % | 80.25 % | 69.68 % |
| | $IPR_{+2tok}$ | 79.15 % | 82.88 % | 82.73 % | 39.54 % | 83.93 % | 73.65 % |
| Python → C++ | *baseline* | 9.05% | 5.36 % | 7.64 % | 6.13 % | 10.58 % | 7.75 % |
| | $IPR_{-2tok}$ | 47.07 % | 46.41 % | 40.63 % | 18.33 % | 37.96 % | 38.08 % |
| | $IPR_{-1tok}$ | 68.40 % | 68.47 % | 62.57 % | 29.06 % | 53.28 % | 56.36 % |
| | IPR | 71.31 % | 72.95 % | 64.94 % | 39.26 % | 58.39 % | 61.37 % |
| | $IPR_{+1tok}$ | 75.62 % | 76.88 % | 71.30 % | 45.15 % | 67.88 % | 67.37 % |
| | $IPR_{+2tok}$ | 75.10 % | 78.02 % | 73.57 % | 48.23 % | 69.71 % | 68.93 % |
| Python → Java | *baseline* | 5.03% | 9.07 % | 7.22% | 1.83 % | 10.55 % | 6.74% |
| | $IPR_{-2tok}$ | 46.07 % | 41.86 % | 39.96 % | 7.07 % | 32.66 % | 33.52 % |
| | $IPR_{-1tok}$ | 65.05 % | 68.21 % | 64.83 % | 11.65 % | 42.21 % | 50.39 % |
| | IPR | 67.09 % | 70.10 % | 66.04 % | 32.46 % | 49.75 % | 57.09 % |
| | $IPR_{+1tok}$ | 70.48 % | 74.32 % | 73.03 % | 20.00 % | 57.79 % | 59.12 % |
| | $IPR_{+2tok}$ | 70.85 % | 73.75 % | 71.96 % | 30.26 % | 62.31 % | 61.83 % |

[*] *baseline*: Randomly select 12 tokens from the pre-target input.

Table 7: The Impact of the Number of High-Information Tokens on Next-Token Prediction in Code Correction Tasks.

| Task | Type | CodeLlama-7b-Instruct | CodeLlama-13b-Instruct | CodeLlama-34b-Instruct | gpt-3.5-turbo | gpt-4-turbo | Ave_1 |
|---|---|---|---|---|---|---|---|
| C++ | *baseline* | 8.05% | 6.61% | 7.79 % | 3.00% | 6.05 % | 6.30 % |
| | $IPR_{-2tok}$ | 76.34 % | 72.98 % | 70.50 % | 30.34 % | 47.67 % | 59.57 % |
| | $IPR_{-1tok}$ | 90.21 % | 90.06 % | 90.50 % | 42.41 % | 69.04 % | 76.44 % |
| | IPR | 91.86 % | 93.70 % | 92.60 % | 56.77 % | 69.69 % | 80.92 % |
| | $IPR_{+1tok}$ | 92.34 % | 94.15 % | 91.54 % | 62.55 % | 80.11 % | 84.14 % |
| | $IPR_{+2tok}$ | 91.05 % | 94.98 % | 91.69 % | 66.41 % | 79.04 % | 84.63 % |
| Java | *baseline* | 7.43% | 7.30% | 9.19 % | 2.39% | 7.93 % | 6.85 % |
| | $IPR_{-2tok}$ | 74.61 % | 73.15 % | 69.95 % | 31.56 % | 48.04 % | 59.46 % |
| | $IPR_{-1tok}$ | 88.92 % | 91.98 % | 90.52 % | 40.91 % | 65.87 % | 75.64 % |
| | IPR | 92.41 % | 94.13 % | 93.00 % | 52.55 % | 72.50 % | 80.92 % |
| | $IPR_{+1tok}$ | 92.57 % | 94.42 % | 92.65 % | 58.35 % | 78.91 % | 83.38 % |
| | $IPR_{+2tok}$ | 92.52 % | 94.07 % | 92.75 % | 63.89 % | 82.08 % | 85.06 % |
| Python | *baseline* | 12.94% | 13.00% | 13.79 % | 3.76% | 9.40 % | 10.58% |
| | $IPR_{-2tok}$ | 79.10 % | 71.90 % | 74.24 % | 37.73 % | 51.20 % | 62.83 % |
| | $IPR_{-1tok}$ | 92.64 % | 92.36 % | 93.10 % | 44.44 % | 69.91 % | 78.49 % |
| | IPR | 92.21 % | 92.79 % | 92.82 % | 54.05 % | 73.25 % | 81.02 % |
| | $IPR_{+1tok}$ | 93.68 % | 93.90 % | 93.62 % | 61.24 % | 82.41 % | 84.97 % |
| | $IPR_{+2tok}$ | 93.73 % | 94.59 % | 93.65 % | 64.36 % | 82.03 % | 85.67 % |

* *baseline*: Randomly select 12 tokens from the pre-target input.

Table 8: The Impact of IPR-Based Code Snippets on Next-Token Prediction in Code Translation Tasks

| Task | Type | CodeLlama-7b-Instruct | CodeLlama-13b-Instruct | CodeLlama-34b-Instruct | gpt-3.5-turbo | gpt-4-turbo | Ave_1 |
|---|---|---|---|---|---|---|---|
| C++ → Java | IPR | 86.43 % | 92.19 % | 92.38 % | 55.33 % | 78.56 % | 80.98 % |
| | $IPR_{remove}$ | 1.99 % | 1.98 % | 2.20 % | 2.91 % | 1.88 % | 2.19 % |
| | $IPR_{corrupt}$ | 3.23 % | 2.97 % | 3.48 % | 2.18 % | 1.23 % | 2.62 % |
| C++ → Python | IPR | 74.01 % | 80.00 % | 73.52 % | 49.15 % | 76.03 % | 70.54 % |
| | $IPR_{remove}$ | 5.75 % | 6.20 % | 6.59 % | 6.82 % | 5.74 % | 6.22 % |
| | $IPR_{corrupt}$ | 5.01 % | 4.23 % | 3.88 % | 3.94 % | 5.08 % | 4.43 % |
| Java → C++ | IPR | 91.41 % | 94.12 % | 92.73 % | 60.27 % | 83.21 % | 84.35 % |
| | $IPR_{remove}$ | 2.19 % | 2.13 % | 2.13 % | 2.97 % | 3.88 % | 2.66 % |
| | $IPR_{corrupt}$ | 2.66 % | 2.65 % | 2.80 % | 1.33 % | 2.02 % | 2.29 % |
| Java → Python | IPR | 70.52 % | 76.54 % | 73.56 % | 39.80 % | 74.70 % | 67.02 % |
| | $IPR_{remove}$ | 4.62 % | 6.68 % | 5.92 % | 7.84 % | 7.74 % | 6.56 % |
| | $IPR_{corrupt}$ | 5.71 % | 3.57 % | 3.59 % | 3.27 % | 4.46 % | 4.12 % |
| Python → C++ | IPR | 71.31 % | 72.95 % | 64.94 % | 39.26 % | 58.39 % | 61.37 % |
| | $IPR_{remove}$ | 3.85 % | 2.90 % | 3.14 % | 5.00 % | 6.27 % | 4.23 % |
| | $IPR_{corrupt}$ | 4.86 % | 6.81 % | 5.91 % | 2.84 % | 3.28 % | 4.74 % |
| Python → Java | IPR | 67.09 % | 70.10 % | 66.04 % | 32.46 % | 49.75 % | 57.09 % |
| | $IPR_{remove}$ | 4.39 % | 4.21 % | 2.99 % | 4.46 % | 2.78 % | 3.77 % |
| | $IPR_{corrupt}$ | 6.27 % | 5.36 % | 2.99 % | 1.32 % | 0.50 % | 3.29 % |

* IPR: IPR-based code snippet= $(x^*_{max\{i-4,1\}:min\{i+4,M\}}, y^*_{max\{j-3,1\}:j-1})$.

* $IPR_{remove}$ represents the code snippets that remove the IPR-based code snippet ⟨src_frag, tgt_frag⟩ from ⟨src_func, tgt_pref⟩.

* $IPR_{corrupt}$ represents the code snippets that corrupt the IPR-based code snippet ⟨src_frag, tgt_frag⟩ from ⟨src_func, tgt_pref⟩.

Table 9: The Impact of IPR-Based Code Snippets on Next-Token Prediction in Code Correction Tasks

| Task | Type | CodeLlama-7b-Instruct | CodeLlama-13b-Instruct | CodeLlama-34b-Instruct | gpt-3.5-turbo | gpt-4-turbo | Ave_1 |
|---|---|---|---|---|---|---|---|
| | IPR | 91.86 % | 93.70 % | 92.60 % | 56.77 % | 69.69 % | 80.92 % |
| C++ | $IPR_{remove}$ | 1.39 % | 2.34 % | 2.08 % | 3.50 % | 2.72 % | 2.41 % |
| | $IPR_{corrupt}$ | 3.05 % | 3.29 % | 3.16 % | 1.82 % | 1.81 % | 2.63 % |
| | IPR | 92.41 % | 94.13 % | 93.00 % | 52.55 % | 72.50 % | 80.92 % |
| Java | $IPR_{remove}$ | 3.14 % | 3.28 % | 3.00 % | 4.61 % | 4.23 % | 3.65 % |
| | $IPR_{corrupt}$ | 2.99 % | 2.92 % | 3.23 % | 2.08 % | 1.74 % | 2.59 % |
| | IPR | 92.21 % | 92.79 % | 92.82 % | 54.05 % | 73.25 % | 81.02 % |
| Python | $IPR_{remove}$ | 2.73 % | 2.64 % | 2.20 % | 3.36 % | 3.45 % | 2.88 % |
| | $IPR_{corrupt}$ | 1.86 % | 1.41 % | 1.47 % | 1.62 % | 2.25 % | 1.72 % |

```python
import openai
openai.api_key = 'API_KEY'

response = openai.ChatCompletion.create(
        model="gpt-4-turbo",
        max_tokens=50,
        messages=[
            {"role": "system",
             "content": "This is a cpp to java translation task."},
            {"role": "user",
             "content": "cpp: int maxSquare ( int b , int m ) { return (\
             b / m - 1 ) * ( b / m ) / 2 ; }"}
        ]
    )

corrected_content = response.choices[0].message.content

print(corrected_content)
```

```
java:
```java
public static int maxSquare(int b, int m) {
    return (b / m - 1) * (b / m) / 2;
}
```
```

(a) Case 1.1: Setting target token = " /"

```python
import openai
openai.api_key = 'API_KEY'

response = openai.ChatCompletion.create(
        model="gpt-4-turbo",
        max_tokens=10,
        messages=[
            {"role": "system",
             "content": "This is a cpp to java translation task.\
                         Please generate the next java token."},
            {"role": "user",
             "content": "cpp: ) * ( b / m ) / 2 java: * ( b"}
        ]
    )

corrected_content = response.choices[0].message.content

print(corrected_content)
```

```
/ m ) / 2
```

(b) Case 1.2: Next generated token = " /" ?    YES

```python
import openai
openai.api_key = 'API_KEY'

response = openai.ChatCompletion.create(
        model="gpt-4-turbo",
        max_tokens=50,
        messages=[
            {"role": "system",
             "content": "This is a cpp to java translation task."},
            {"role": "user",
             "content": "cpp: long long int findSum ( int n ) { return n *\
             ( n + 1 ) * ( n + 2 ) * ( 3 * n + 1 ) / 24 ; }"}
        ]
    )

corrected_content = response.choices[0].message.content

print(corrected_content)
```

```
java:
```java
public class SumCalculator {
    public static long findSum(int n) {
        return n * (n + 1) * (n + 2) * (3 * n + 1) / 24
```
```

(c) Case 2.1: Setting target token = " * "

```python
import openai
openai.api_key = 'API_KEY'

response = openai.ChatCompletion.create(
        model="gpt-4-turbo",
        max_tokens=10,
        messages=[
            {"role": "system",
             "content": "This is a cpp to java translation task.\
                         Please generate the next java token."},
            {"role": "user",
             "content": "cpp: ) * ( 3 * n + 1 ) java: * (3"}
        ]
    )

corrected_content = response.choices[0].message.content

print(corrected_content)
```

```
* n + 1)
```

(d) Case 2.2: Next generated token = " * " ?    YES

```python
import openai
openai.api_key = 'API_KEY'

response = openai.ChatCompletion.create(
        model="gpt-4-turbo",
        max_tokens=50,
        messages=[
            {"role": "system",
             "content": "This is a cpp to java translation task."},
            {"role": "user",
             "content": "cpp: int power ( int n ) { if ( n == 1 ) return\
             2 ; return 2 * power ( n - 1 ) ; }"}
        ]
    )

corrected_content = response.choices[0].message.content

print(corrected_content)
```

```
java:
```java
int power(int n) {
    if (n == 1) return 2;
    return 2 * power(n - 1);
}
```
```

(e) Case 3.1: Setting target token = "power"

```python
import openai
openai.api_key = 'API_KEY'

response = openai.ChatCompletion.create(
        model="gpt-4-turbo",
        max_tokens=10,
        messages=[
            {"role": "system",
             "content": "This is a cpp to java translation task.\
                         Please generate the next java token."},
            {"role": "user",
             "content": "cpp: ; return 2 * power ( n - 1 java: return 2 *"}
        ]
    )

corrected_content = response.choices[0].message.content

print(corrected_content)
```

```
power(n - 1
```

(f) Case 3.2: Next generated token = "power" ?    YES

```python
import openai
openai.api_key = 'API_KEY'

response = openai.ChatCompletion.create(
        model="gpt-4-turbo",
        max_tokens=50,
        messages=[
            {"role": "system",
             "content": "This is a cpp to java translation task."},
            {"role": "user",
             "content": "cpp: int multiplyWith3Point5 ( int x ) { return\
             ( x << 1 ) + x + ( x >> 1 ) ; } }"}
        ]
    )

corrected_content = response.choices[0].message.content

print(corrected_content)
```

```
Here's the translation of the provided C++ function into a Java method:

```java
public int multiplyWith3Point5(int x) {
    return (x << 1) + x + (x >> 1);
}
```
```

(g) Case 4.1: Setting target token = ">>"

```python
import openai
openai.api_key = 'API_KEY'

response = openai.ChatCompletion.create(
        model="gpt-4-turbo",
        max_tokens=10,
        messages=[
            {"role": "system",
             "content": "This is a cpp to java translation task.\
                         Please generate the next java token."},
            {"role": "user",
             "content": "cpp: x + ( x >> 1 ) ; } java: + ( x"}
        ]
    )

corrected_content = response.choices[0].message.content

print(corrected_content)
```

```
>> 1 ) ; }
```

(h) Case 4.2: Next generated token = ">>" ?    YES

Figure 6: **Examples of IPR-based Code Snippets for Next-Token Prediction.** In the left sub-figures, we randomly select a token in tgt_pref as the target token and extract the corresponding IPR-based code snippet. In right subfigures, we attempt to predict the next token using the IPR-based code snippets. Furthermore, since gpt-3.5/4-turbo can only be accessed via the API, we employ prompt engineering by including the instruction "*Please generate the next java token.*" in the prompt to guide the model to generate the next token directly.

Table 10: Java-Python Language Pair and Its Unit Test Cases Example

| Type | Details |
| --- | --- |
| Source Java function | ```java
public static int getLineSize(int x1, int x2, int y1, int y2) {
    if (x1 == x2) {
        return Math.abs(y1 - y2);
    } else {
        return Math.abs(x1 - x2);
    }
}
``` |
| Python function | ```python
def get_line_size(x1, x2, y1, y2):
    if x1 == x2:
        return abs(y1 - y2)
    else:
        return abs(x1 - x2)
``` |
| Java unit tests | ```java
/*
 * This file was automatically generated by EvoSuite
 * Wed Dec 07 08:44:05 GMT 2022
 */

import org.junit.Test;
import static org.junit.Assert.*;
import org.evosuite.runtime.EvoRunner;
import org.evosuite.runtime.EvoRunnerParameters;
import org.junit.runner.RunWith;

@RunWith(EvoRunner.class)
@EvoRunnerParameters(mockJVMNonDeterminism = true, useVFS = true, useVNET = true,
        resetStaticState = true, separateClassLoader = true)
public class CLASS_a59109abfc5d_ESTest extends CLASS_a59109abfc5d_ESTest_scaffolding {

  @Test(timeout = 4000)
  public void test0() throws Throwable {
      int int0 = CLASS_a59109abfc5d.getLineSize(0, 0, (-11228), (-26867));
      assertEquals(15639, int0);
  }

  @Test(timeout = 4000)
  public void test1() throws Throwable {
      int int0 = CLASS_a59109abfc5d.getLineSize(14804, (-1), 1, 113128);
      assertEquals(14805, int0);
  }

  @Test(timeout = 4000)
  public void test2() throws Throwable {
      int int0 = CLASS_a59109abfc5d.getLineSize(1, 1, 1, 1);
      assertEquals(0, int0);
  }

  @Test(timeout = 4000)
  public void test3() throws Throwable {
      int int0 = CLASS_a59109abfc5d.getLineSize(0, 48313, 0, 12019);
      assertEquals(48313, int0);
  }

  @Test(timeout = 4000)
  public void test4() throws Throwable {
      CLASS_a59109abfc5d cLASS_a59109abfc5d_0 = new CLASS_a59109abfc5d();
  }}
``` |
| Python unit tests | ```python
import numpy as np
import math
from math import *
import collections
from collections import *
import heapq
import itertools
import random
import sys
import unittest

#TOFILL
class CLASS_a59109abfc5d(unittest.TestCase):

  def test0(self):
      int0 = f_filled(0, 0, (-11228), (-26867))
      assert 15639 == int0

  def test1(self):
      int0 = f_filled(14804, (-1), 1, 113128)
      assert 14805 == int0

  def test2(self):
      int0 = f_filled(1, 1, 1, 1)
      assert 0 == int0

  def test3(self):
      int0 = f_filled(0, 48313, 0, 12019)
      assert 48313 == int0

if __name__ == '__main__':
    unittest.main()
``` |

Table 11: **Examples of Similarities Between the Internal Representations of Pre-Target Inputs and Code Snippets Surrounding Core Tokens.**

| Type | Similarity | ⟨src_func, tgt_pref⟩ and ⟨src_frag, tgt_frag⟩ |
|---|---|---|
| C++ → Java | 0.7746 | **[EX.1]:** ⟨ ' double mul ( const int a , double b ) { return ( a ) * b ; } ', 'public static double mul ( final int a , final double b ) { return (' ⟩ → a |
| | | **[EX.1]\*:** ⟨ ' ) { return ( a ) * b ; ', '{ return (' ⟩ → a |
| | 0.7083 | **[EX.2]:** ⟨ ' int sum ( int * a ) { int result = 0 ; for ( int i = 0 ; i < sizeof ( a ) / sizeof ( int ) ; i ++ ) result += a [ i ] ; return result ; } ', 'public static int sum ( int [ ] a ) { int result = 0 ; for (' ⟩ → int |
| | | **[EX.2]\*:** ⟨ ' 0 ; for ( int i = 0 ; ', '; for (' ⟩ → int |
| | 0.7007 | **[EX.3]:** ⟨ ' int clamp ( int x , int a , int b ) { if ( a > b ) return x ; if ( x < a ) return a ; else if ( x > b ) return b ; else return x ; } ', 'public static int clamp ( int x , int a , int b ) { if ( a > b ) return x ; if ( x < a ) return a ;' ⟩ → else |
| | | **[EX.3]\*:** ⟨ ' return a ; else if ( x > ', 'return a ;' ⟩ → else |
| | 0.7880 | **[EX.4]:** ⟨ ' bool gt ( double number , double actual ) { return actual > number ; } ', 'public static boolean greaterThan ( double number , double actual ) { return' ⟩ → actual |
| | | **[EX.4]\*:** ⟨ ' actual ) { return actual > number ; } ', ') { return' ⟩ → actual |
| C++ → Python | 0.7781 | **[EX.1]:** ⟨ ' int max ( int a , int b ) { return a > b ? a : b ; } ', 'def gt ( a , b ) : return a if a' ⟩ → > |
| | | **[EX.1]\*:** ⟨ ' ) { return a > b ? a : ', 'a if a' ⟩ → > |
| | 0.7454 | **[EX.2]:** ⟨ ' int sign ( int b0 ) { return ( b0 < 0 ) ? - 1 : ( b0 > 0 ) ? 1 : 0 ; } ', 'def _sign ( b0 ) : return np .' ⟩ → sign |
| | | **[EX.2]\*:** ⟨ ' b0 < 0 ) ? - 1 : ( ', 'return np .' ⟩ → sign |
| | 0.8218 | **[EX.3]:** ⟨ ' double distance ( double * p1 , double * p2 ) { double sum = 0 ; for ( int i = 0 ; i < sizeof ( * p1 ) / sizeof ( * p1 ) ; i ++ ) { const double dp = * p1 - * p2 ; sum += dp * dp ; } return sqrt ( sum ) ; } ', 'def distance ( p1 , p2 ) : sum = 0 for i in range ( len ( p1 ) ) : dp = p1 [ i ] - p2 [ i ] sum += dp * dp return math .' ⟩ → sqrt |
| | | **[EX.3]\*:** ⟨ ' dp ; } return sqrt ( sum ) ; ', 'return math .' ⟩ → sqrt |
| | 0.7109 | **[EX.4]:** ⟨ ' float trunc ( float number , int precision ) { return float ( floor ( number * pow ( 10 , precision ) ) / pow ( 10 , precision ) ) ; } ', 'def trunc ( number , prec ) : return float ( math .' ⟩ → floor |
| | | **[EX.4]\*:** ⟨ ' { return float ( floor ( number * pow ', '( math .' ⟩ → floor |
| Java → C++ | 0.7856 | **[EX.1]:** ⟨ ' public static double mul ( final int a , final double b ) { return ( a ) * b ; } ', 'double mul ( const int a , double b ) { return ( a' ⟩ → ) |
| | | **[EX.1]\*:** ⟨ ' { return ( a ) * b ; } ', 'return ( a' ⟩ → ) |
| | 0.8468 | **[EX.2]:** ⟨ ' public static int lerp ( int a , int b , float value ) { return int ( a + ( b - a ) * value ) ; } ', 'int lerp ( int a , int b , float value ) { return int ( a + ( b' ⟩ → - |
| | | **[EX.2]\*:** ⟨ ' ( a + ( b - a ) * value ', 'a + ( b' ⟩ → - |
| | 0.7672 | **[EX.3]:** ⟨ ' public static boolean greaterThan ( double number , double actual ) { return actual > number ; } ', 'bool gt ( double number , double actual ) { return actual' ⟩ → > |
| | | **[EX.3]\*:** ⟨ ' ) { return actual > number ; } ', '{ return actual' ⟩ → > |
| | 0.7356 | **[EX.4]:** ⟨ ' public static int test ( ) { int h = 1 ; { } int j = 2 ; return 120 + j + h ; } ', 'int test ( ) { int h = 1 ; { }' ⟩ → int |
| | | **[EX.4]\*:** ⟨ ' 1 ; { } int j = 2 ; ', '; { }' ⟩ → int |
| Python → Java | 0.7183 | **[EX.1]:** ⟨ ' def clamp ( val , min , max ) : return min if val < min else max if val > max else val ', 'public static final int clamp ( int val , int min , int max ) { return ( val' ⟩ → < |
| | | **[EX.1]\*:** ⟨ ' return min if val < min else max if ', 'return ( val' ⟩ → < |
| | 0.7946 | **[EX.2]:** ⟨ ' def pad ( c ) : if c >= 10 : return str ( c ) else : return '0' + str ( c ) ', 'public static String pad ( int c ) { if ( c' ⟩ → >= |
| | | **[EX.2]\*:** ⟨ ' if c >= 10 : return str ', 'if ( c' ⟩ → >= |
| | 0.8168 | **[EX.3]:** ⟨ ' def Min ( a , b ) : return min ( a , b ) ', 'public static final int min ( int a , int b ) { return Math .' ⟩ → min |
| | | **[EX.3]\*:** ⟨ ' : return min ( a , b ', 'return Math .' ⟩ → min |
| | 0.8244 | **[EX.4]:** ⟨ ' def distance ( x1 , y1 , x2 , y2 ) : return math . sqrt ( ( pow ( x1 - x2 , 2 ) + pow ( y1 - y2 , 2 ) ) ) ', 'public static double distance ( double x1 , double y1 , double x2 , double y2 ) { return Math .' ⟩ → sqrt |
| | | **[EX.4]\*:** ⟨ ' return math . sqrt ( ( pow ( ', 'return Math .' ⟩ → sqrt |
| Java → Python | 0.8580 | **[EX.1]:** ⟨ ' public static final int min ( int a , int b ) { return Math . min ( a , b ) ; } ', 'def Min ( a , b ) : return min' ⟩ → ( |
| | | **[EX.1]\*:** ⟨ ' return Math . min ( a , b ) ', ' return min' ⟩ → ( |
| | 0.8681 | **[EX.2]:** ⟨ ' public static final int clamp ( int val , int min , int max ) { return ( val < min ) ? min : ( val > max ) ? max : val ; } ', 'def clamp ( val , min , max ) : return min if val' ⟩ → < |
| | | **[EX.2]\*:** ⟨ ' { return ( val < min ) ? min ', 'min if val' ⟩ → < |
| | 0.7808 | **[EX.3]:** ⟨ ' public static String right ( String s , int count ) { if ( s == null ) { return null ; } count = s . length ( ) - count ; return s . substring ( ( count < 0 ) ? 0 : ( count < s . length ( ) ) ? count : s . length ( ) ) ; } ', 'def right ( s , count ) : if s is' ⟩ → None |
| | | **[EX.3]\*:** ⟨ ' if ( s == null ) { return null ', 'if s is' ⟩ → None |
| | 0.8413 | **[EX.4]:** ⟨ ' public static double distance ( double x1 , double y1 , double x2 , double y2 ) { return Math . sqrt ( ( Math . pow ( x1 - x2 , 2 ) + Math . pow ( y1 - y2 , 2 ) ) ) ; } ', 'def distance ( x1 , y1 , x2 , y2 ) : return math .' ⟩ → sqrt |
| | | **[EX.4]\*:** ⟨ ' { return Math . sqrt ( ( Math . ', 'return math .' ⟩ → sqrt |
| Python → Java | 0.7135 | **[EX.1]:** ⟨ ' def dst ( x1 , y1 , z1 , x2 , y2 , z2 ) : a = x2 - x1 b = y2 - y1 c = z2 - z1 return float ( math . sqrt ( a * a + b * b + c * c ) ) ', 'float dst ( float x1 , float y1 , float z1 , float x2 , float y2 , float z2 ) { float a = x2 - x1 ; float b = y2 - y1 ; float c = z2 - z1 ; return ( float ) sqrt ( a' ⟩ → * |
| | | **[EX.1]\*:** ⟨ ' . sqrt ( a * a + b * ', 'sqrt ( a' ⟩ → * |
| | 0.7773 | **[EX.2]:** ⟨ ' def gt ( a , b ) : return a if a > b else b ', 'int max ( int a , int b ) { return a' ⟩ → > |
| | | **[EX.2]\*:** ⟨ ' return a if a > b else b ', '{ return a' ⟩ → > |
| | 0.8405 | **[EX.3]:** ⟨ ' def trunc ( number , prec ) : return float ( math . floor ( number * pow ( 10 , prec ) ) / pow ( 10 , prec ) ) ', 'float trunc ( float number , int precision ) { return float ( floor ( number *' ⟩ → pow |
| | | **[EX.3]\*:** ⟨ ' . floor ( number * pow ( 10 , prec ', '( number *' ⟩ → pow |
| | 0.7745 | **[EX.4]:** ⟨ ' def odd ( _ ) : if 0 > _ : return math . floor ( ( _ - 1 ) / 2 ) * 2 + 1 else : return math . ceil ( ( _ + 1 ) / 2 ) * 2 - 1 ', 'double odd ( double number ) { if ( 0 ) return' ⟩ → floor |
| | | **[EX.4]\*:** ⟨ ' return math . floor ( ( _ - ', '0 ) return' ⟩ → floor |

* ⟨·, ·⟩ → '\*' : ⟨·, ·⟩: It represents the decision-making basis for each generated token '\*' in the corrected Python function.
* [EX_i]: It represents the process of ⟨src_func, tgt_pref⟩ → target token.
* [EX_i]\*: It represents the process of ⟨src_frag, tgt_frag⟩ → target token.
* C: It represents the corrupted token.

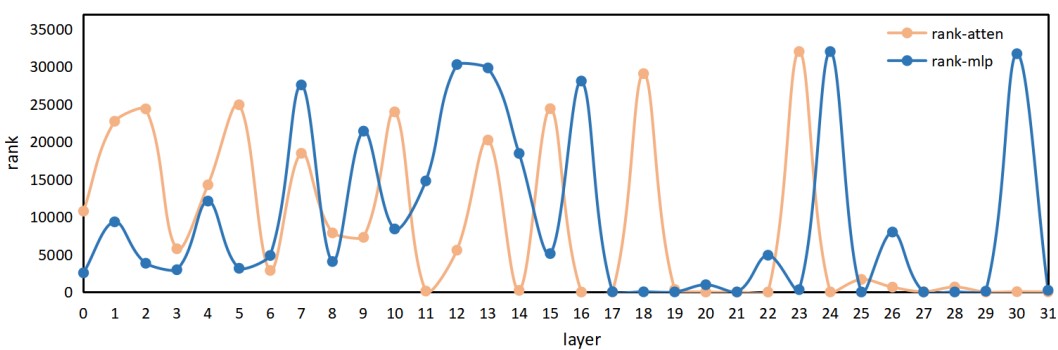

Figure 7: Target Token Ranking Across Layers in CodeLlama-7b-Instruct via Logit Lens.

Table 12: **Detailed Sample Size of IPR-Based Code Snippets for Next-Token Prediction in Code Translation Task.** It includes the evaluation of various models, including CodeLlama-7b-Instruct, CodeLlama-13b-Instruct, CodeLlama-34b-Instruct, gpt-3.5-turbo, and gpt-4-turbo, in the context of code translation. The evaluation encompasses interactive translations among three programming languages: Java, Python, and C++.

| | CL-7b-Instruct | | CL-13b-Instruct | | CL-34b-Instruct | | gpt-3.5-turbo | | gpt-4-turbo | |
| --- | --- | --- | --- | --- | --- | --- | --- | --- | --- | --- |
| | succ | total | succ | total | succ | total | succ | total | succ | total |
| C++ → Java | 1032 | 1194 | 1298 | 1408 | 1383 | 1497 | 374 | 676 | 535 | 681 |
| C++ → Python | 393 | 531 | 480 | 600 | 372 | 506 | 174 | 354 | 314 | 413 |
| Java → C++ | 1372 | 1501 | 1457 | 1548 | 1390 | 1499 | 584 | 969 | 783 | 941 |
| Java → Python | 311 | 441 | 336 | 439 | 345 | 469 | 121 | 304 | 251 | 336 |
| Python → C++ | 343 | 481 | 418 | 573 | 350 | 539 | 106 | 270 | 160 | 274 |
| Python → Java | 210 | 313 | 354 | 505 | 350 | 530 | 74 | 228 | 99 | 199 |

Table 13: **Detailed Sample Size of IPR-Based Code Snippets for Next-Token Prediction in Code Correction Task.** It provides details of the test sets used for CodeLlama-7b-Instruct, CodeLlama-13b-Instruct, CodeLlama-34b-Instruct, gpt-3.5-turbo, and gpt-4-turbo, specifically focusing on code correction tasks in Java, Python, and C++.

| | CL-7b-Instruct | | CL-13b-Instruct | | CL-34b-Instruct | | gpt-3.5-turbo | | gpt-4-turbo | |
| --- | --- | --- | --- | --- | --- | --- | --- | --- | --- | --- |
| | succ | total | succ | total | succ | total | succ | total | succ | total |
| C++ | 1366 | 1487 | 1472 | 1571 | 1363 | 1472 | 482 | 849 | 538 | 772 |
| Java | 1194 | 1292 | 1283 | 1363 | 1196 | 1286 | 433 | 824 | 667 | 920 |
| Python | 1479 | 1604 | 1505 | 1622 | 1513 | 1630 | 467 | 864 | 619 | 845 |

Table 14: **Detailed Sample Size of IPR-Based Code Snippets for Next-Token Prediction in Code Completion Task.** There are two types of code snippets: i) a 3-token snippet $\{y_{j-3}, y_{j-2}, y_{j-1}\}$, which focuses on a few tokens preceding the target token, and ii) an 8-token snippet $\{y_{j-8}, \ldots, y_{j-1}\}$, which incorporates a broader context from the prefix.

| | CL-7b-Instruct | | CL-13b-Instruct | | CL-34b-Instruct | | gpt-3.5-turbo | | gpt-4-turbo | |
| --- | --- | --- | --- | --- | --- | --- | --- | --- | --- | --- |
| | succ | total | succ | total | succ | total | succ | total | succ | total |
| C++_tok3 | 262 | 966 | 272 | 956 | 259 | 966 | 69 | 251 | 55 | 243 |
| Java_tok3 | 306 | 900 | 251 | 900 | 254 | 871 | 62 | 277 | 74 | 260 |
| Python_tok3 | 245 | 874 | 250 | 888 | 290 | 959 | 49 | 228 | 78 | 252 |
| C++_tok8 | 646 | 1022 | 643 | 1015 | 628 | 1004 | 116 | 284 | 120 | 275 |
| Java_tok8 | 639 | 956 | 599 | 967 | 581 | 928 | 109 | 282 | 123 | 278 |
| Python_tok8 | 460 | 915 | 454 | 894 | 507 | 988 | 88 | 232 | 119 | 266 |

