# OpenReview forum: "A Consistent Pattern for Identifying Decisive Code Snippets for LLM-Based Code Inference"
_ICLR.cc/2025/Conference — ICLR 2025 Conference Withdrawn Submission_

### Official Review · Reviewer_XX9J · 2024-10-29

**Soundness:** 2
**Presentation:** 2
**Contribution:** 3
**Rating:** 6
**Confidence:** 3

**Summary:**

This paper investigates which parts of pre-target inputs significantly influence next-token prediction in programming languages within LLMs, including CodeLlama and GPT. The authors employ the causal tracing method and propose a multi-phase causal tracing process to analyze the importance distribution pattern of high-information tokens, demonstrating that specific code snippets at certain locations play a critical role in inference and exhibit a consistent pattern, named the Important Position Rule (IPR). Finally, experiments are conducted across various code downstream tasks to validate their findings.

**Strengths:**

1. This paper explores a very interesting question: which parts of pre-target inputs are most influential for next-token prediction in the context of programming languages?
2. Based on the causal tracing method, the authors propose the IPR method to evaluate the influence between model tokens.
3. The models used in the experiments cover a range of sizes and are applied to various coding tasks for validation.

**Weaknesses:**

There are some symbols, such as what s1 and s2 in Figure 2 are not defined, and the same applies to Stage 4 of Figure 1.
Additionally, there should be clear conclusions to answer the two key questions raised in lines 356-360.

**Questions:**

1. In line 242, the source of the "3,650 samples" used for single-token perturbations needs clarification.
2. In section 4.1, the specific construction steps of the Test Dataset require further elaboration, ideally with detailed examples or illustrations, and it should be noted whether the dataset is publicly available.
3. Additionally, does the CAUSAL TRACING METHOD only consider the influence of a single token on the target token, without accounting for the effects of multiple or consecutive tokens?
4. In line 180, it mentions that specific tokens come from "the output of the final layer of a Transformer before the linear layer." However, given that ChatGPT is a black-box model accessed via an API, how to obtain the token output from the ChatGPT model is a question that needs to be addressed.
5. Note that this is not essential but could be considered: in the APPLICATION: IPR-BASED KNOWLEDGE EDITING section, providing a comparison with prefix tuning techniques, such as LoRA and prefix tuning, including the number of parameters.

---

> ### Author Response · Authors · 2024-11-18
> **Response to Reviewer XX9J**
>
> Thanks very much for your valuable comments!  Based on the issues you pointed out, we have added the relevant experiments. In addition, we would like to reiterate the contributions of this paper. The research background(objective), main work, and main contributions are outlined as follows.
>
> Background:\
> The syntactic structures inherent in natural language, <subject, relation, object>, provide a solid knowledge support for tasks like interpretable inference and knowledge editing.  However, it remains unclear whether similar mechanisms exist in code-based LLM research. This gap presents challenges  for the aforementioned  research in programming language contexts.  In this paper, we demonstrate that code snippets following important position rule play a crucial role in guiding LLM-based code inference, which provides a knowledge support for extending the above research tasks in the code field.
>
> Main work:\
> (1) In this paper, we propose the causal tracing method and the multi-phase causal tracing process, revealing that the importance distribution of high-information tokens follows a consistent pattern, named the Important Position Rule (IPR). (2) To evaluate the rationality and feasibility of IPR, we first validate the performance of IPR-based code snippets across multi-task, multi-language, and multi-model scenarios. Then, we assess the performance of high-information snippets with different lengths, as well as corrupting/removing the IPR-based snippet in the pre-target input on the next token prediction. (3) Finally, we combine IPR with ROME in the context of programming languages, generalizing this knowledge editing beyond the NLP context.
>
> Contribution:\
> (1) In this paper, we first propose a causal tracing method that can identify high-information tokens in both the input sequence and the generated output prefix, which makes up for the neglect of tgt\_pref in previous tracing methods. \
> (2)we propose a multi-phase causal tracing process to analyze the importance distribution pattern of high-information tokens.\
> (3) We reveal a  consistent pattern of high-information tokens, named the important position rule. It provides knowledge support for the application of knowledge editing and interpretable inference, effectively filling a gap in previous research due to the lack of knowledge support. \
> (4)we successfully combine IPR with the knowledge editing method ROME. To our knowledge, this is the first application of knowledge editing in the context of programming languages.\
> (5)We verified the generalisation and feasibility of IPR in multi-task, multi-language, and multi-model scenarios. In addition, we also measured different lengths of IPR-based snippets and performed ablation experiments for further validation.

---

> ### Author Response · Authors · 2024-11-19
> **Response to Reviewer XX9J**
>
> \
> Response to Weaknesses:\
> Thanks for your suggestion. We have made some changes in the paper. For the two questions in Section 4, in multi-task, multi-language and multi-model scenarios, we respectively verified whether the original target token can be successfully predicted using only IPR-based code snippet. In Section 4.2, we give the results on the two problems. In addition, the symbols in Figure 1 can be found in Section 5.1.
>
> Response to Question_1:\
> We show the detailed generation process in APPENDIX A.1.2.
>
> Response to Question_2:\
> (1) We show the generation process of all datasets used in this paper in APPENDIX A.1. In this process, we introduce the source databases, literature, and data volume used in the data collection process in detail. In addition, we offer detailed data on the length distribution of the dataset, and compare it with benchmark datasets such as HumanEval and MBPP  in Table 5 and Figure 5.\
> (2) For extracting and using IPR-based code snippets for next token prediction, we provide an example in Figure 5.
>
> Response to Question_3:\
> This paper considers the impact of a single token and multiple consecutive tokens on next-token prediction. In addition, we have added relevant experiments based on your suggestion.\
> (1) The impact of a single token: In this paper, the causal tracing method is used to identify a single token in the input sequence and the generated output prefix that highly contributes to the next-token prediction.\
> (2) The impact of consecutive tokens: In Section 4, we use IPR to extract consecutive high-information tokens (i.e., IPR-based code snippet) for next-token prediction. We verify it in a multi-task, multi-language, and multi-model scenario, and the robust results illustrate the impact of multiple consecutive tokens on next-token prediction.\
> (3) The impact of consecutive tokens (new): According to your suggestion, in order to further verify the impact of multiple consecutive tokens on next-token prediction, we have added relevant experiments.\
> i) In APPENDIX A.2, we verify the impact of different ranges of consecutive code snippets on next-token prediction (see Tables 6 and 7 for results).\
> ii) In APPENDIXA.3, we added related ablation experiments. We corrupt or remove the IPR-based consecutive tokens in the original pre-target input, thereby analyzing the impact of multiple consecutive tokens on next-token prediction (results in Tables 8 and 9).\
> iii) In Table 11, we provide some examples of similarity comparisons between consecutive high-information snippets and the original pre-target input.
>
> Response to Question_4:\
> (1) There may be some misunderstandings. "the output of the final layer of a Transformer before the linear layer" is the internal representation of <src\_func, tgt\_pref>. The causal tracing method uses it to track high-information tokens. \
> (2) If I understand correctly, you mean how to directly extract the next generated next token of <src\_frag, tgt\_frag> for gpt-3.5/4-turbo. we leverage prompt engineering by including the instruction, ``Please generate the next java token.'' in the prompt to guide the model to directly generate the next token. (Some examples are shown in Figure 6.)
>
> Response to Question_5:\
> Thanks very much for your suggestions! Our latest work is to further improve the application of this paper. We will provide relevant experiments in future work.

---

> > ### Comment · Reviewer_XX9J · 2024-11-23
> > **Thanks for the reply.**
> >
> > I appreciate the additional experiments and clarifications addressing some of the questions. However, I have new concerns regarding the example in Figure 6, particularly with Case 1. For example,
> > 1. The selection criteria for the target tokens were not explained.
> > 2. Why were both GPT-3.5 and GPT-4.0 used simultaneously?
> > 3. As suggested by other reviewers, the lack of baseline comparisons is notable. If the target tokens were selected randomly or according to other rules, how would the next token prediction perform in such cases?

---

> ### Author Response · Authors · 2024-11-23
> **Response to Reviewer XX9J**
>
> We really appreciate your reply!\
> Response to Question_1:\
> We explain the definition of the target token in Section 3.1.(In simple terms, any token in tgt\_pref can be selected as a target token. )
>
> $Definition$: In the code correction scenario, a potentially incorrect source code  $x$  (referred to as src\_func) is mapped to a corrected code  $y$ through LLM. Both $x$ and $y$ are functions in the considered programming language like Python, and each is represented by a sequence of tokens  $x= [ x_1, x_2, \dots, x_M ]$ and $y=[ y_1,  y_2, \dots, y_N ]$. \
> Consider an autoregressive Transformer language model, where all previously generated tokens $y_{1:j-1}$  are treated as additional input when generating the next token $y_{j}$, where $j\in [1, N]$ .  We designate $y_j$ as a ``$target$  $token$'' and  $y_{1:j-1}$  as  tgt\_pref (i.e. a ''$target$  $prefix$''). For a target token  $y_j$, since the output distribution $p(y_j|x,y_{1:j-1})$ is conditioned on both the  src\_func $x$ and the tgt\_pref   $y_{1:j-1}$, we define the pre-target input as  <src\_func, tgt\_pref>.
>
> Response to Question_2:\
> We are very sorry for the confusion caused by the wrong figures! We have resubmitted the new revision. Sorry again!
>
> Response to Question_3:\
> We have fully supplemented the experiments based on suggestions from other reviewers. \
> (1) In our work, any token in tgt\_pref can be selected as a target token. Below is the relationship between target token,  <src\_func, tgt\_pref> and <src\_frag, tgt\_frag>.\
> (i) The generation of the target token depends on both src_func and tgt_pref. This paper focuses on the single generated token, aiming to identify which tokens in <src\_func, tgt\_pref> highly contribute to the next generated token (i.e., target token). \
> (ii) We extract high-information code snippets <src\_frag, tgt\_frag> form the <src\_func, tgt\_pref> based on important position rule (IPR), called IPR-based code snippets (i.e., decisive code snippets).
>
> (2) We understand your concerns, but there are some misunderstandings here. Other reviewers suggested that we randomly select tokens from <src\_func, tgt\_pref> as "decisive" snippets or try code snippets of different lengths to illustrate the feasibility of IPR. We added experiments to analyse the impact of next token prediction by trying code snippets of different lengths and by corrupting the IPR-based snippet in the original pre-target input.\
> (i) In APPENDIX A.2, we verify the impact of code snippets with different ranges on next token prediction  across multi-task, multi-language, and multi-model scenarios (see Tables 6 and 7).\
> (ii) In APPENDIX A.3, we conducted ablation experiments across multi-task, multi-language, and multi-model scenarios. We found that when we removed/corrputed the IPR-based snippet in the complete pre-target input <src\_func, tgt\_pref>, the success rate was very low, no more than 8% in all scenarios. Even in most scenarios, the performance of next token prediction of damaged pre-target input does not exceed 3.5%, which can effectively prove the role of IPR in pre-target input <src\_func, tgt\_pref> (results in Tables 8 and 9).

---

> > ### Comment · Reviewer_XX9J · 2024-11-23
> >
> > I still have some concerns regarding Figure 6. The example in Figure 6 demonstrates that core code segments + prompts can effectively predict the next token when completing a code snippet. However,
> > 1. If the same GPT API is used, the contextual relationship within the model might increase the consistency of token predictions when using key code segments.
> > 2. Even if this next-token prediction is effective, its practical significance for real-world tasks seems questionable (as inferred from the example in Figure 6).

---

> ### Author Response · Authors · 2024-11-23
> **Response to Reviewer XX9J**
>
> We fully understand your question as your question is the background of our paper.  Regarding your two questions, in our paper, IPR main purpose, LLM inference principles, the experimental design, dataset generation, motivation and application all reflect the answers to your questions. We will briefly answer the two questions from the above aspects.
>
> (1-1)The main purpose of IPR is to explore the internal contextual relationship of the model. Our work is to explore the knowledge source of the next generated token when the model performs code inference. We found that when the model performs code inference, the next generated token mainly depends on the IPR-based snippet in <src\_func, tgt\_pref>. Meanwhile, the IPR demonstrates excellent generalization across various LLMs, tasks, and multilingual scenarios.
>
> (1-2) As you mentioned before, we apologize again for the wrong figures uploaded previously, we have updated the figures in last reply. Most importantly, we must use the same model because different models interpret context and preference in different ways, resulting in different tgt_pref outputs for the same src_func.  (see Section 3.1, Section 4.1 and APPENDIX A.1)\
> (i) For example, in gpt-4-turbo, the src_func serves as the input, and its output is tgt_pref. Within gpt-4-turbo, for any <src_func, tgt_pref> → target token, we aim to know whether the high-information tokens that determines the target token in <src_func, tgt_pref> follows IPR.  Therefore, we conducted various experiments using IPR-based code snippets to perform next-token prediction.\
> (ii) It is worth noting that, in experiments (i.e., Section 4, APPENDIX A.2, A.3), we extracted the IPR-based code snippet only to verify the role of IPR across multi-task, multi-language, and multi-model scenarios, not the application of IPR(see IPR application in Section 5.)
>
> (1-3) Based on the above, we have written about why we use IPR-based snippets for prediction and how to build a  <src\_frag, tgt\_frag> dataset in Section 4 and APPENDIX A.1. (We have excerpted some content.) \
> To validate that IPR-based  code snippets are decisive for LLM-based code inference, we consider  the following two questions: Q1: Can only code snippets identified  by IPR reliably predict the target token? Q2: Does IPR exhibit strong generalization, making it applicable across diverse programming languages, tasks, and LLMs?  In this experiments, it is necessary to build a dedicated IPR-based snippet dataset for each model. The following is the process of constructing the dataset in the experiment.\
> (i)First, we construct specialized <src\_func, tgt\_pref> dataset for CodeLlama-7b-Instruct, CodeLlama-13b-Instruct, CodeLlama-34b-Instruct, gpt-3.5-turbo, and gpt-4-turbo, targeting tasks such as code translation, code correction, and code completion.\
> (ii)Based on this, we respectively construct specialized IPR-based snippet dataset for CodeLlama-7b-Instruct, CodeLlama-13b-Instruct, CodeLlama-34b-Instruct, gpt-3.5-turbo, and gpt-4-turbo, targeting tasks such as code translation, code correction, and code completion.\
> (Figure 6 is just to show the IPR-based code snippet to the readers. )
>
> (2-1) The significance of IPR:\
> The syntactic structures inherent in natural language, <subject, relation, object>, provide a solid knowledge support for tasks like interpretable inference and knowledge editing. However, it remains unclear whether similar mechanisms exist in code-based LLM research. This gap presents challenges for the aforementioned research in programming language contexts. In this paper, we demonstrate that code snippets following important position rule (IPR) play a crucial role in guiding LLM-based code inference, which provides a knowledge support for extending the above research tasks in the code field.
>
> (2-2) The practical significance of IPR:\
> Regarding the application of IPR, we combine IPR with ROME in the context of programming languages, generalizing this knowledge editing beyond the NLP context (see Section 5). In addition, in our ongoing work, we have extended the application work in the code field and achieved good results.
>
> PS: It is worth noting that, in experiments (i.e., Section 4, APPENDIX A.2, A.3), we extracted the IPR-based code snippet only to verify the role of IPR across multi-task, multi-language, and multi-model scenarios, not the application of IPR (See Application of IPR in Section 5).
>
>
> MOST IMPORTANTLY, we really value your suggestions and hope to hear from you! Your feedback is our motivation for future work! Thank you!

---

> > ### Comment · Reviewer_XX9J · 2024-11-25
> >
> > I appreciate the additional experiments and explanations provided by the authors, and I will change my score to a positive one. However, I still believe that the advantages brought by the IPR technique in the application of NLP tasks need to be clarified.

---

> ### Author Response · Authors · 2024-11-25
> **To Reviewer XX9J**
>
> Thank you very much for your suggestions and encouragement! Your suggestions are the motivation for our future work.
>
> The main goal of this paper is to reveal the consistent pattern of high-information tokens in programming language contexts. We have carefully validated this across various languages, tasks, and models. To showcase its practical application, we included an example of applying the Important Position Rule (IPR) in the domain of knowledge editing. Additionally, we will release the code, models, and datasets in the future. Our ongoing work extends the exploration of IPR in knowledge editing, and we look forward to sharing the results soon!

---

### Official Review · Reviewer_PzAg · 2024-11-02

**Soundness:** 2
**Presentation:** 1
**Contribution:** 2
**Rating:** 3
**Confidence:** 4

**Summary:**

This paper investigates what tokens within input sequences critically influence the decoding of target tokens in code language models. Unlike prior methods that focus on perturbing tokens solely within the source sequence, this paper introduces the idea of incorporating the prefix of the target sequence for correlation analysis. The results demonstrate that next-token prediction relies heavily on core tokens (tokens that exhibit analogous functionality and purpose) in the provided sequence. The authors further validate this approach across three downstream tasks—code correlation, code completion, and code translation. Additionally, the method is applied to knowledge editing, yielding promising results in repairing translation errors.

**Strengths:**

• The focus on interpreting code language models is valuable and timely. This paper contributes a new idea by incorporating target sequence prefixes in correlation analysis.

• The experiments cover multiple coding tasks, including code translation, code completion, and code correlation, adding depth to the evaluation.

**Weaknesses:**

•	The novelty is somewhat limited, as the proposed approach bears similarity to prior methods (e.g., arxiv:1908.04626, arXiv:1612.08220,arXiv: 2004.14786) except including the decoding prefix for analysis.

•	The study only examines correlations between individual tokens, neglecting potential dependencies across phrases or combinations of tokens at multiple positions, which is common in structured source code (e.g., bracket pairs).

•	The evaluation lacks comparisons to baselines. For instance, Tables 1 and 2 present results only across variants of LLMs. Comparisons to traditional NLP methods, particularly those excluding the target prefix, would strengthen the evaluation. The following baseline methods may be considered for compassion: arxiv:1908.04626, arXiv:1612.08220,arXiv: 2004.14786

•	Ablation studies are needed to assess factors such as the number of core tokens, and the positional impact of target tokens within the sequence. For example, the authors can vary the number of core tokens from 1 to 5 and evaluate the effect on performance.

•	While source code inherently has structured elements, this paper overlooks such structural information. Without it, the potential <subject, relation, object> analogies found in natural language are not fully analogous to source code.

•	The practical utility of the code correlation task is questionable.

•	Some essential details are missing. For example, in Line 373, the authors mentioned that they constructed a specialized IPR-based dataset. However, there are no details about how the authors constructed the evaluation dataset, such as the source of the code samples, the criteria for selection, and any preprocessing steps. In particular, I wonder why not use existing benchmarks for these tasks such as HumanEval and CodeSearchNet.

**Questions:**

1.	Why were only the 4th, 8th, 12th, and 17th positions in the target sequence chosen for information detection? Why specifically select seven pre-target inputs?

2.	The IPR method assumes the presence of core tokens influencing the target token. What happens if no core tokens exist?

3.	How did you construct the evaluation dataset? Why not use existing benchmarks?

---

> ### Author Response · Authors · 2024-11-21
> **Response to Reviewer PzAg (Part-1)**
>
> Thanks for your suggestions! However, I believe there are some misunderstandings. As indicated by the title, "A Consistent Pattern for Identifying Decisive Code Snippets for LLM-Based Code Inference",  the core of this paper is not the causal tracing method, but the discovery that, across various LLMs, the importance distribution of high-information tokens for the next generated token follows a consistent pattern —— Importance Pattern Rule (IPR). In other words, the core of this paper is IPR, a  consistent pattern. Furthermore, we verified the generalizability and feasibility of IPR in downstream tasks,  instead of causal tracing method.  Finally, we apply IPR to knowledge editing, rather than causal tracing method. \
>  (We briefly introduce the background and workflow below. Contributions can be found in "Response to Weaknesses_1".)
>
> Background:\
> The syntactic structures inherent in natural language, <subject, relation, object>, provide a solid knowledge support for tasks like interpretable inference and knowledge editing. However, it remains unclear whether similar mechanisms exist in code-based LLM research. This gap presents challenges for the aforementioned research in programming language contexts. In this paper, we demonstrate that code snippets following important position rule play a crucial role in guiding LLM-based code inference, which provides a knowledge support for extending the above research tasks in the code field.
>
> Workflow:\
> (1) In this paper, we propose the causal tracing method and the multi-phase causal tracing process, revealing that the importance distribution of high-information tokens follows a consistent pattern, named the Important Position Rule (IPR). (2) To evaluate the generalizability and feasibility of IPR, we first validate the performance of IPR-based code snippets across multi-task, multi-language, and multi-model scenarios. Then, we assess the performance of high-information snippets with different lengths and the impact of corrupting/removing IPR-based snippet in the pre-target input on the next token prediction.  (3) Finally, we combine IPR with ROME in the  context of programming languages, generalizing this knowledge editing beyond the NLP context.

---

> ### Author Response · Authors · 2024-11-21
> **Response to Reviewer PzAg (Part-2)**
>
> Response to Weaknesses_1:\
> Thanks for your feedback. However, there are some misunderstandings. The core of this paper is IPR, a consistent pattern (see paper title), rather than causal tracing method. $The$ $following$ $are$ $the$ $main$ $contributions$.\
> (1) In this paper, we first propose a causal tracing method that can identify high-information tokens in both the input sequence and the generated output prefix, which makes up for the neglect of tgt_pref in previous tracing methods. In addition,we propose a multi-phase causal tracing process to analyze the importance distribution pattern of high-information tokens.\
> (2) We reveal a  consistent pattern of high-information tokens, named the important position rule(IPR). It provides knowledge support for the application of knowledge editing and interpretable inference, effectively filling a gap in previous research due to the lack of knowledge support.\
> (3) we successfully combine IPR with the knowledge editing method ROME. To our knowledge, this is the first application of knowledge editing in the context of programming languages.\
> (4) We verified the generalisation of IPR in multi-task, multi-language, and multi-model scenarios. In addition, we also measured different lengths of IPR-based snippets and performed ablation experiments for further validation.
>
> Response to Weaknesses_2:\
> This paper considers the impact of both a single token and multiple consecutive tokens on next-token prediction. In addition, we have added relevant experiments based on your suggestion.\
> (1) Impact of a single token: In this paper, the causal tracing method is used to identify a single token in the input sequence and the generated output prefix that highly contributes to the next-token prediction.\
> (2) Impact of consecutive tokens: In Section 4, we use IPR to extract consecutive high-information tokens (i.e., IPR-based code snippet) for next-token prediction. We verify it in a multi-task, multi-language, and multi-model scenario, and the robust results illustrate the impact of multiple consecutive tokens on next-token prediction.\
> (3) Impact of consecutive tokens (new): According to your suggestion, in order to further verify the impact of multiple consecutive tokens on next-token prediction, we added relevant experiments.\
> (i) In APPENDIX A.2, we verify the impact of consecutive code snippets with different lengths on next-token prediction (see Tables 6 and 7).\
> (ii) In APPENDIX A.3, we added related ablation experiments. We corrupt or remove the IPR-based consecutive tokens in the original pre-target input, thereby analyzing the impact of multiple consecutive tokens on next-token prediction (see Tables 8 and 9).
>
> Response to Weaknesses_3:\
> (1)Thanks for your suggestion! But there are some misunderstandings here. In our experiments, as shown in Tables 1 and 2, we tested IPR, instead of the causal tracing method. If we understand correctly, you mean to add a baseline for the experiment, i.e. for the IPR. For this, following Reviewer 8mN2's suggestion, we added the baseline: "random selection of tokens as 'decisive'" as a basic baseline comparison. We put the results in Tables 6 and 7.\
> (2) There are fundamental differences between our causal tracing approach and existing tracing methods.\
> (i) Existing methods focus on the complete output sequence, identifying only tokens in the input sequence (i.e., src\_func) that have a high impact on the entire output sequence.\
> (ii) In contrast, our causal tracing method focuses on the single generated token, aiming to identify which tokens in <src\_func, tgt\_pref> highly contribute to the next generated token. (This is because the generation of the target token depends on both src\_func and tgt\_pref, so it is absolutely necessary to consider tgt\_pref in our work.)\
> (iii) Thanks for your paper recommendation! We discuss these papers in "Related Work".
>
> Response to Weaknesses_4:\
> Based on your suggestion, we added related experiments.\
> (1) In APPENDIX A.3, we added ablation experiments, where we corrupt or remove IPR-based code snippets from the original pre-target input, thereby analyzing the impact on next token prediction(see Tables 8 and 9).\
> (2) In APPENDIX A.2, we added experiments to discuss the impact of code snippets of different lengths on next-token prediction. (see Tables 6 and 7).\
> (3) Furthermore, when constructing the IPR-based snippet dataset, we considered target tokens from different positions, as well as input sequences of different lengths.\
> (4) In addition, the number of core tokens is immutable. As shown in definitions 1 and 2, the core token in src\_func is functionally equivalent to the target token, and the core token in tgt\_pref is the preceding token of the target token. If we understand correctly, you mean trying code snippets of different lengths centered around the core token (See (2) for related experiments).

---

> ### Author Response · Authors · 2024-11-21
> **Response to Reviewer PzAg  (Part-3)**
>
> Response to Weaknesses_5:\
> (1) We did not ignore structural information. In our data processing, the AutoTokenizer in the code-based LLM treats symbols such as "{" and "}" as tokens.\
> (2) The core token in src_frag exhibits analogous functionality and purpose to the target token, similar to the relationship between “subject” and “object”. In this paper, we regard the core token in src_frag as the "subject" and the target token as the "object".
>
> Response to Weaknesses_6:\
> Thank you for your feedback. However, there are some misunderstandings. We apply IPR to knowledge editing rather than causal tracing method. We mentioned it in the Abstract, Introduction & Contribution, and Section 5.\
> (1) We provide a large number of experiments on the generalizability and feasibility of IPR. First, we validate the performance of IPR-based code snippets across multi-language, multi-task and multi-model scenarios. To further assess the rationality of IPR, we evaluate the performance of high-information snippets with different lengths, as well as corrupting/removing the IPR-based snippet in the pre-target input on the next token prediction.\
> (2) Regarding the application of IPR in knowledge editing scenarios, we provide experimental results, theoretical foundations, and a detailed data generation process (See APPENDIX A.1 for details on dataset generation.). In addition, our latest work is an extension of the application part, and we are confident in the role of IPR in the field of knowledge editing.
>
> Response to Weaknesses_7:\
> (1) Thanks for your suggestions! In APPENDIX A.1, for all datasets used in this paper, we provide the detailed generation process, source database, data volume and length distribution of the dataset.\
> (2) IPR-based snippet datasets consist of high-information tokens from both input sequences and generated prefixes as well as target tokens. However, CodeSearchNet and HumanEval only contain the input sequence src\_func, which is not suitable for our experiments.
>
> Response to Questions_1:\
> (1) We propose a multi-phase causal tracing process to analyze the importance distribution pattern of high-information tokens. This process consists of three phases: (A)Observation of the distribution pattern of high-information tokens in small samples; (B)Assessment of whether high-information tokens exhibit clustering phenomena in large samples; and (C)Evaluation of cluster centers for high-information tokens in large samples.\
> (i) Phase A is just a preliminary discovery process. In the small samples (seven samples), we found that for target tokens at different positions \{4, 8, 12, 17\}, the clustering of high-information tokens presents a certain pattern. Here, {4, 8, 12, 17} is just to represent the target tokens at different positions, which are chosen randomly. The seven small samples are also randomly selected. (The main purpose of starting with a small sample case is to make it easier for readers to understand the subsequent work.)\
> (ii) Phases B and C are further exploration processes. In the large sample, we perform a comprehensive analysis of the importance distribution of high-information tokens for target tokens in different positions (This means that different lengths of tgt_pref).
>
> Response to Questions_2:\
> (1) Thanks for your suggestion. However, there are some misunderstandings here. As shown in definitions 1 and 2, we define two core tokens in <src_func, tgt_pref>. The core token in src_func is functionally equivalent to the target token, and the core token in tgt_pref is the preceding token of the target token. Since the core token in tgt_pref will always exist, there is no situation where the core token cannot be found. If we understand correctly, you mean the situation where the core token does not exist in src_func.\
> (2) In natural language scenarios, researchers usually focus on the triple <subject, relation, object> for knowledge editing or other related research. Therefore, in our study, we also consider the similar scenario where the core token exists in src_func.\
> (3) Moreover, because we are very interested in this question, we conducted a simple exploration and obtained interesting results. In the field of code completion, since the output prefix is a continuation of the input sequence, there is no core token in src_func. Based on this, we tried to extract only the IPR-based snippet in tgt_pref (including only 3 tokens) for next token prediction. Surprisingly, we found that we could still achieve a success rate of approximately 30%, as shown in Table 3.
>
> Response to Questions_3:\
> See "Response to Weaknesses_7" for the answer.

---

### Official Review · Reviewer_8mN2 · 2024-11-04

**Soundness:** 2
**Presentation:** 3
**Contribution:** 3
**Rating:** 6
**Confidence:** 3

**Summary:**

The authors propose a causal tracing method to identify high information content tokens. This method involves corrupting one prefix token and comparing the similarity between the high-dimensional predicted next token and the original next token. A lower similarity indicates higher information content. Building on the causal tracing method, the authors conduct experiments from small to large sample sizes and discover a consistent pattern in the distribution of high information content tokens, which they name the Important Position Rule (IPR). The authors combine IPR with the knowledge editing method ROME in code translation tasks to demonstrate one application of IPR, resulting in an increase in the correction rate from 62.73% to 75.31%.

**Strengths:**

- The identification of high information content tokens of code is insightful.
- Clear progression from theoretical framework to practical applications.
- The paper is well-written and easy to follow.
- The proposed method is effective in knowledge editing tasks.

**Weaknesses:**

There are several major weaknesses in this paper:

1. This paper lacks justification for the datasets used. Where are the 3650 samples in Section 3.2 from?  What are the sources, dataset sizes and lengths of the correct and incorrect codes in Section 4.1? And how did the authors collect the 25227 Java functions with 2975 incorrect Python translations in Section 5.2? Where are the test cases mentioned in Line 493 from? The authors should provide more details about the datasets used in the experiments.
2. Many key decisions in this paper are unclear. For example, how did you determine the "4-token radius" for sec and "2-token radius" for the target in Line 343? Are there any supporting quantitative analysis, or are they just visual inspection from Figure 4? How will they affect the results in Section 4 and 5? Furthermore, how did you decide on keeping "12 high-information tokens" in the experiments in Section 4.2? What is the average length of the original input? How will the results change if you keep more or fewer high-information tokens? The authors should provide more details about the key parameter choices in the paper.
3. The idea of identifying high-information tokens has also been explored in prompt compression methods like LLMLingua [1] and Selective Context [2]. Do you think it is necessary to discuss these works and compare them in the paper?
4. There is a lack of baselines in the experiments. Other than the prompt compression methods mentioned above, some simple baselines like random selection of tokens as "decisive" should at least be included in the experiments to verify the effectiveness of the proposed method.
5. The methodology for calculating the success rate in Section 4.2 is ambiguous. For translation and completion tasks that typically involve generating a sequence of tokens, how exactly is the success rate computed? Is it based on token-level accuracy or sequence-level exact match? If the evaluation is done on a token-by-token basis, how are the test samples constructed? Does each test case involve predicting only a single token? It would be helpful to clarify the details in the paper.
6. Upon seeing the title of this paper, I've been curious to know what kind of snippets would be decisive. However, this question remains unclear after reading the paper. What does the "decisive code snippets" identified in this paper look like in practice? Are they some specific code patterns or structures, or is it just position-based? It would be helpful to provide some examples in the paper.

And some minor issues:

1. There are some concerns about scalability to real-world code. From Figure 2, all examples appear to have source lengths of around 30 tokens or fewer. However, production code is typically much longer. Can the proposed method be generalized to identify high information content tokens in longer code snippets?
2. In Tables 5-7 of the appendix, why do different tasks have varying sample sizes when using the same task? For example, in Table 5, the C++ to Python task has 529 samples for CL-7b-Instruct and 604 for CL-13b-Instruct.
3. In Line 481-482, The KL divergence term $D_{KL}(P_G[x|p'] || P_G[x|p'])$ computes divergence between identical distributions, I guess the correct version should be $D_{KL}(P_G[x|p'] || P_G[x|p])$. And the optimization term "argmin_z" uses an undefined variable z, will "argmin_v" be more appropriate?
4. Including examples in the appendix that show the next token predictions before and after using IPR would be helpful to better understand the method. Do the improvements in predictions primarily relate to syntax or semantics?
5. What is the exact version of GPT-4-turbo used in your experiments? There are multiple versions of GPT-4-turbo available from OpenAI according to [3], so specifying the version would help ensure reproducibility.

[1] LLMLingua: Compressing Prompts for Accelerated Inference of Large Language Models, EMNLP 2023

[2] Compressing Context to Enhance Inference Efficiency of Large Language Models, ACL 2023

[3] https://platform.openai.com/docs/models/o1#gpt-4-turbo-and-gpt-4

**Questions:**

Please address the concerns in the "Weaknesses" section.

---

> ### Author Response · Authors · 2024-11-18
> **Response to Reviewer 8mN2**
>
> Thank you for providing such high-quality feedback on our work. We also appreciate you pointing out the issues in the paper. Following your suggestion, we add a number of relevant experiments and analyses to the APPENDIX of the paper.
>
> Response to Weaknesses_1: \
> (1) In APPENDIX A.1, for all datasets used in this paper, we provide the detailed generation process and data volume.\
> (2) We present the code length distribution of our dataset, comparing it with benchmark datasets such as HumanEval and MBPP. (See Table 5 and Figure 5)\
> (3) Additionally, we present examples from the Java-Python unit test dataset in Table 10, and provide examples of IPR-based code snippets in Figure 6.
>
> Response to Weaknesses_2:\
> (1) Thanks for your suggestions on experimental settings. We have added relevant experiments as you suggested. In APPENDIX A.2, we discussed the impact of different lengths of code snippets on next-token prediction (results in Tables 6 and 7).\
> (2) Regarding the length of the codes used in the paper, we provide detailed data in Figure 5 and Table 5, and compare them with benchmark datasets such as HumanEval and MBPP.
>
> Response to Weaknesses_3:\
> We are indeed interested in the paper you suggested, which is related to our ongoing work and discussed in the related work of this paper. However, the focus of the above work and this paper is different. This paper only focuses on which tokens in <src\_func, tgt\_pref> have a high contribution to the target token, with an emphasis on the next single token prediction. Furthermore, the generation of the target token depends on both src\_func and tgt\_pref. Therefore, it is necessary to consider tgt_pref.
>
> Response to Weaknesses_4:\
> (1) Based on your suggestion, we verify the impact of code snippets with different ranges on next token prediction in APPENDIX A.2 (results in Tables 6 and 7).\
> (2) Then, we conducted ablation experiments in APPENDIX A.3. We found that when we removed/corrputed the IPR-based snippet in the complete pre-target input <src\_func, tgt\_pref>, the success rate was very low, no more than 8% in all scenarios . Even in most scenarios, the success rate of damaged pre-target input does not exceed 3.5%, (results in Tables 8 and 9).
>
> Response to Weaknesses_5:\
> (1) The calculation of success rate is based on token-level. We use the IPR-based code snippets as inputs to the LLM, comparing the next generated token $y_j^{'}$ with the original target token $y_j$. (In Section 4.1, we provide more details.)\
> (2) Different models interpret context and preferences in distinct ways, leading to varying tgt\_pref outputs for the same src\_func.  Therefore,  it is necessary to construct dedicated IPR-based snippet datasets for each model. Based on this, we respectively construct specialized IPR-based snippet dataset for CodeLlama-7b-Instruct, CodeLlama-13b-Instruct, CodeLlama-34b-Instruct, gpt-3.5-turbo, and gpt-4-turbo, targeting tasks such as code translation, code correction, and code completion. (we provide technical details in APPENDIX A.1.1.)
>
> Response to Weaknesses_6:\
> (1) Sorry for the confusion. The decisive code snippet is the IPR-based code snippet.  (See Section 3.4)\
> Given a target token $y_j$, high-information tokens cluster within a 4-token radius around the core token $x_i^*$ in  src\_func  and within a 2-token radius around the core token $y_{j-1}^*$ in tgt\_pref. This pattern can be expressed as $(x_{max\{i-4,1\}: min\{i+4,M\}}^*,y_{max\{j-3,1\}: j-1}^*)$, known as the Important Position Rule (IPR).
>  In LLM-based code inference, code snippets identified by the importance position rule  are decisive for next-token prediction.\
> (2) In addition, we provide examples in Figure 6.
>
> Response to minor_issue_1:\
> Yes, the proposed method be generalized to identify high information content tokens in longer code snippets. We present the length distribution of the input sequence src\_func in Section 4.2 in Figure 5 and Table 5.
>
> Response to minor_issue_2:\
> Different models interpret context and preferences in distinct ways, leading to varying tgt\_pref outputs for the same src\_func.  Therefore,  we construct specialized IPR-based snippet datasets for each model. Additionally, we need to extract the code in an automated manner, but some of the output formats do not conform, which leads to certain differences.
>
> Response to minor_issue_3:\
> Thanks for your correction,  the changes have been made.
>
> Response to minor_issue_4:\
> (1) We added relevant experiment in APPENDIX A.2 (results in Tables 6 and 7).\
> (2) I think the improvements in predictions are primarily due to semantics. High-information tokens in src\_func are typically clustered around the core token, which exhibits functional equivalence to the target token.\
> (3) We provide examples in Figure 6.
>
> Response to minor_issue_5:\
> The following is the gpt version.\
> gpt-3.5-turbo: https://platform.openai.com/docs/models#gpt-3-5-turbo  \
> gpt-4-turbo: https://platform.openai.com/docs/models#gpt-4-turbo-and-gpt-4

---

> > ### Comment · Reviewer_8mN2 · 2024-11-24
> > **Thanks for the reply and the enriched experiments.**
> >
> > Thank you for your detailed response to my review. I appreciate the additional experiments and analyses you have added to address my concerns! I've been waiting for the update of the following major concern I had:
> >
> > > > 4. Other than the prompt compression methods mentioned above, some simple baselines like random selection of tokens as "decisive" should at least be included in the experiments to verify the effectiveness of the proposed method.
> > >
> > > Response to Weaknesses_4:
> > > (1) Based on your suggestion, we verify the impact of code snippets with different ranges on next token prediction in APPENDIX A.2 (results in Tables 6 and 7).
> > > (2) Then, we conducted ablation experiments in APPENDIX A.3. We found that when we removed/corrputed the IPR-based snippet in the complete pre-target input <src_func, tgt_pref>, the success rate was very low, no more than 8% in all scenarios . Even in most scenarios, the success rate of damaged pre-target input does not exceed 3.5%, (results in Tables 8 and 9).
> >
> > I appreciate the new ablation experiments in Appendix A.3 showing the impact of removing/corrupting IPR-based snippets. However, I want to clarify that my original concern about baselines was not fully addressed. Specifically, I suggested including "random selection of tokens as 'decisive'" as a basic baseline comparison, instead of selecting IPR-based tokens, to verify its effectiveness.

---

> ### Author Response · Authors · 2024-11-25
> **To Reviewer 8mN2:  We added the baseline.**
>
> Thank you for your suggestion!  According to your suggestion, we added the baseline: "random selection of tokens as 'decisive'" as a basic baseline comparison. We put the results in Tables 6 and 7.
>
> If possible, we'd really like to get your feedback. Thanks again!!

---

> > ### Comment · Reviewer_8mN2 · 2024-11-25
> >
> > Thank you for your follow-up result. The revision addresses my concern. I will further update my score.

---

> > > ### Author Response · Authors · 2024-11-25
> > >
> > > We truly appreciate the time and effort you have dedicated to providing valuable feedback. Your insightful comments and suggestions have greatly enhanced the quality of our work, and we are grateful for your expertise and support throughout this process.

---

### Author Response · Authors · 2024-11-30
**Concern regarding the reviewer PzAg's feedback**

Dear Reviewers and Meta-Reviewers,\
We appreciate Reviewer PzAg’s detailed feedback. However, we believe that the Reviewer PzAg has a significant misunderstanding of the main work and test object of our paper, which we would like to clarify.  In particular, the Reviewer PzAg misunderstands the object of the "Experiment" and "Application" sections. \
Specifically, the Reviewer PzAg seems to view the object of the "Experiment" and "Application" sections as the causal tracing method. In contrast, the actual object of the "Experiment" and "Application" sections is a consistent pattern, namely the IPR. To emphasize this, we mention IPR 177 times throughout the paper, and the title itself—"A Consistent Pattern for Identifying Decisive Code Snippets for LLM-Based Code Inference"—reflects this central concept. To further illustrate our concerns, we provide the following excerpts from Reviewer PzAg's feedback.

>$Summary$ (from Reviewer PzAg):\
(i) Unlike prior methods that focus on perturbing tokens solely within the source sequence, this paper introduces the idea of incorporating the prefix of the target sequence for correlation analysis. \
(ii) The authors further validate $this$ $approach$ across three downstream tasks—code correlation, code completion, and code translation. \
(iii) Additionally, $the$ $method$ is applied to knowledge editing, yielding promising results in repairing translation errors.

>$Strengths$ (from Reviewer PzAg):\
(i) The focus on interpreting code language models is valuable and timely. This paper contributes a new idea by incorporating target sequence prefixes in correlation analysis.\
(ii) The experiments cover multiple coding tasks, including code translation, code completion, and code correlation, adding depth to the evaluation.

>$Weaknesses$ (from Reviewer PzAg): \
(i) The novelty is somewhat limited, as the proposed approach bears similarity to prior methods (e.g., arxiv:1908.04626, arXiv:1612.08220,arXiv: 2004.14786) except including the decoding prefix for analysis.\
(ii) The evaluation lacks comparisons to baselines. For instance, Tables 1 and 2 present results only across variants of LLMs. Comparisons to traditional NLP methods, particularly those excluding the target prefix, would strengthen the evaluation. The following baseline methods may be considered for compassion: arxiv:1908.04626, arXiv:1612.08220,arXiv: 2004.14786

In the "$Summary$" section, Reviewer PzAg appears to have misunderstood the object of our "Experiment" and "Application" sections, describing it as a causal tracing method rather than IPR. Furthermore, the "$Strengths$" section indicates that the reviewer may not have recognized IPR as the central concept of our paper. Finally, in the "$Weaknesses$" section,  the Reviewer PzAg believes that Tables 1 and 2 are results related to the causal tracing method, and that the innovation of this paper is the causal tracing method. This further suggests that reviewer PzAg does not seem to understand what story we are telling and what we are testing.

In fact, the core of our paper is IPR, a consistent pattern. The object of our"Experiment" and "Application" sections is IPR, and the results presented in Tables 1 and 2 are tests of IPR.  We would like to emphasize that IPR is mentioned 177 times in the paper, and the title explicitly highlights it: "A Consistent Pattern for Identifying Decisive Code Snippets for LLM-Based Code Inference."

**Most importantly, due to Reviewer PzAg's misunderstanding of the object of the 'Experiment' and 'Application' sections, we infer that Reviewer PzAg may not fully understand the core of the paper, despite it being explicitly stated in the title, abstract, contributions, and mentioned 177 times throughout the paper. As a result, 5 out of the 10 suggestions provided by Reviewer PzAg are either completely or partially unreasonable. (We provided detailed clarification in our response to Reviewer PzAg, but received no feedback.)**


Authors

---

### Author Response · Authors · 2024-12-03
**General Response**

Dear Reviewers and Meta-Reviewers,\
We would like to express our sincere gratitude to all the reviewers for their insightful questions and suggestions. In response, we have revised the paper accordingly and have uploaded the updated PDF version. The following are the motivation and main contributions of this paper:

**Motivation:**\
The syntactic structures inherent in natural language, <subject, relation, object>, provide a solid knowledge support for tasks like interpretable inference and knowledge editing. However, it remains unclear whether similar mechanisms exist in code-based LLM research. This gap presents challenges for the aforementioned research in programming language contexts. \
In this paper, we demonstrate that code snippets following important position rule (IPR) play a crucial role in guiding LLM-based code inference, which provides a knowledge support for extending the above research tasks in the code field.\
 (We provide a detailed explanation of the discovery, validation, and application of the IPR, and extensively test its generalization across diverse tasks, models, and programming languages.)

**Main Contributions:**\
(1) In this paper, we first propose a causal tracing method that identifies high-information tokens in both the input sequence and the generated output prefix, making up for the neglect of tgt_pref in previous tracing methods.\
(2) Through a multi-phase causal tracing process, we reveal a consistent pattern of high-information tokens, named the important position rule (IPR). It provides knowledge support for the application of knowledge editing and interpretable inference, effectively filling a gap in previous research due to the lack of knowledge support.\
(3) We successfully combine IPR with the knowledge editing method ROME, extending knowledge editing beyond the NLP context. To our knowledge, this is the first application of knowledge editing in the context of programming languages.\
(4) We validate the generalization of IPR in code inference across diverse models, tasks, and programming languages, including code translation, code correction, and code completion, utilizing CodeLlama-7b/13b/34b-Instruct and gpt-3.5/4-turbo with Java, Python, and C++. Our evaluation demonstrates that code snippets identifed by IPR play a critical role in next-token prediction, providing valuable interpretability for LLM-based code inference.


 Authors

---

### Note · Authors · 2025-01-27

I have read and agree with the venue's withdrawal policy on behalf of myself and my co-authors.